

# A new criterion for determining the representative elementary volume of translucent porous media and inner contaminant

Ming Wu[1, 2], Jianfeng Wu[2*], Jichun Wu[2], and Bill X. Hu[1*]

[1] Institute of Groundwater and Earth Sciences, Jinan University, Guangzhou 510632, China

[2] Key Laboratory of Surficial Geochemistry, Ministry of Education; Department of Hydrosciences, School of Earth Sciences and Engineering, Nanjing University, Nanjing 210023, China

*Correspondence to*: J.F. Wu (jfwu@nju.edu.cn), B.X. Hu (billhu@jnu.edu.cn)



**ABSTRACT**

Representative elementary volume (REV) is essential to measure and quantify the effective parameters of a complex heterogeneous medium. Since previous REV estimation criteria having multiple limitations, a new criterion ($\chi^i$) is proposed to estimate REV of a translucent material based on light transmission techniques. Two sandbox experiments are performed to estimate REVs of porosity, density, tortuosity and perchloroethylene (PCE) plume using multiple REV estimation criteria. In comparison with $\chi^i$, previous REV estimation criteria based on the coefficient of variation ($C_V^i$), the entropy dimension ($DI^i$) and the relative gradient error ($\varepsilon_g^i$) are tested in REV quantification of translucent silica and inner PCE plume to achieve their corresponding effects. Results suggest that new criterion ($\chi^i$) can effectively identify the REV in the materials, whereas the coefficient of variation ($C_V^i$) and entropy dimension ($DI^i$) cannot. The relative gradient error ($\varepsilon_g^i$) can make the REV plateau obvious, while random fluctuations make the REV plateau uneasy to identify accurately. Therefore, the new criterion is appropriate for REV estimation for the translucent materials and inner contaminant. Models are built based on Gaussian equation to simulate the distribution of REVs for media properties, which frequency of REV is dense in the middle and sparse on both sides. REV estimation of PCE plume indicates high level of porosity lead to large value of mean and standard deviation for REVs of PCE saturation ($S_o$) and PCE-water interfacial area ($A_{OW}$). Fitted equations are derived for distribution of REVs for PCE plume related to $d_m$ (distances from mass center to considered point) and $d_I$ (distances from injection



position to considered point). Moreover, relationships between REVs of PCE plume and
$S_o$ are fitted using regression analysis. Results suggest a decreasing trend appears for
$S_o$-REV when $S_o$ increases, while $A_{ow}$-REV increases with increasing of $S_o$.
**Keywords:** new criterion; representative elementary volume (REV);translucent material
**1. Introduction**

Modelling groundwater and contaminant (such as hazardous irons) transport in

subsurface environment is based on the premise that micro-structure of aquifer exist a
representative elementary volume (REV) (Wang et al., 2016; Lei and Shi, 2019). REV act
as a micro-scale characteristic, which is important to improve our understanding of
materials, inner fluid flow and other processes (Brown and Hsieh, 2000;
Costanza-Robinson et al., 2011; Wu et al., 2017). Previous studies suggested that even the
Platinum-Nanoparticle-Catalyzed hydrogenation reactions and ion transport through
angstrom-scale slits in cell activity existed apparent size effect, implying size effect is a
wide characteristic for many process and materials (Bai et al., 2016; Esfandiar et al., 2017).
With the help of REV, a porous medium can be treated as continuum medium (Brown and
Hsieh, 2000; Kang et al., 2003; Müller and Siegesmund, 2010; Teruel and Rizwan-uddin,
2010; Hendrick et al., 2012; Wang et al., 2012; Ukrainczyk and Koenders, 2014; Kim and
Mohanty, 2016; Gilevska et al., 2019). A conceptual representation of "REV curve",
characterizing porosity ($n$) change with measured scale ($L$) increment, is presented in Fig.
1c. Based on the characteristic of REV curve in Fig. 1c, the REV curve can be divided into
three regions. When measured scale is in region I, the porosity fluctuates drastically at
small scales. As measured scale size ranging between $L_{min}$ and $L_{max}$, a flat plateau with



constant and steady value is encountered and the property is factored into its average value.
Material property in spatial scales less than $L_{min}$ is spatially varied portions with small
scale, which can be easily influenced by individual pores in micro-structure such as region
I (Fig. 1c). Likewise, material property is allowed drift to new values in spatial scale
above $L_{max}$ due to additional morphological structures of large field heterogeneity (region
III). As a matter of fact, REV scale of region II can be derived between the small and
spatially varied property in region I and large field variability in region III. However, the
lower and upper boundaries $L_{min}$ and $L_{max}$ of REV plateau is hard to be identified in reality
(Brown and Hsieh, 2000; Costanza-Robinson et al., 2011).
As technology advanced and progressed, non-destructive and non-invasive
techniques of x-ray and gamma ray micro-tomography were utilized for micro-structure
characteristic measurement in materials  (Ghilardi, 1993; Brown and Hsieh, 2000;
Niemet and Selker, 2001; Bob et al., 2008; Al-Raoush and Papadopoulos, 2010;
Costanza-Robinson et al., 2011; Al-Raoush, 2012; Borges and Pires, 2012; Fernandes et
al., 2012; Rozenbaum and Roscoat, 2014; Pereira Nunes et al., 2016; Piccoli et al., 2019).
Generally, REV estimation also was usually implemented by micro visualization and
scanning of X-ray and gramma ray in laboratory (Brown and Hsieh, 2000; Razavi   et al.,
2007; Nordahl   and   Ringrose,   2008;   Al-Raoush   and   Papadopoulos,   2010;
Costanza-Robinson et al., 2011; Rozenbaum and Roscoat, 2014; Borges et al., 2018),
while different criteria were utilized to quantify REV (Brown and Hsieh, 2000; Martínez
et al., 2007; Nordahl and Ringrose, 2008; Costanza-Robinson et al., 2011). Lower
boundary scale $L_{min}$ of REV was identified by means of entropy dimension ( $DI^{i}$ ) for eight





soil samples (Martínez et al., 2007). Further, REV scale of permeability for ripple
laminated sandstone intercalated with mudstone was estimated using the coefficient of
variation ($C_V^i$), which the REV scale is identified by the variability among the ten
samples to achieve average REV scale (Nordahl and Ringrose, 2008). As a result, only
one REV boundary was identified and not every sample can be estimated effectively
(Nordahl and Ringrose, 2008). More interestingly, REV scales for porosity, moisture
saturation and air-water interfacial areas in porous media were estimated by a criterion
named the relative gradient error ($\varepsilon_g^i$)(Costanza-Robinson et al., 2011). In summary, the
REV estimation was made by multiple kinds of criteria, while the REV identification
effects of these criteria were not clear.

In this study, a new criterion ($\chi^i$) for REV estimation is proposed to identify the REV

scale of the translucent silica and inner contaminant. Two perchloroethylene (PCE)
transport experiments are conducted in two dimensional (2D) sandboxes to test the effect
of different REV estimation criteria. Translucent silica is selected for associated REV
analysis due to its extensive utilization in laboratory experiment of exploring groundwater
flow and contaminant migration behavior in micro-structure of a sandy aquifer (Niemet
and Selker, 2001; Bob et al., 2008; Costanza-Robinson et al., 2011). Moreover, translucent
silica is also an important material applied in numerous industries (Bouvry et al., 2016). In
laboratory experiments, translucent silica is packed in 2D sandboxes where porosity,
density, tortuosity, PCE saturation are derived by light transmission technique (Fig. 1a).
Porosity and PCE saturation are selected as the representative variables to explore
corresponding REV estimation by different criteria, which is very essential and significant



for REV identification. Previous criteria such as the coefficient of variation ($C_V^i$),
entropy dimension ($DI^i$), the relative gradient error ($\varepsilon_g^i$) and the new criterion-$\chi^i$ are
tested in REV estimation. Associated effects are analyzed to achieve the best criterion of
effective and appropriate quantification of REV.
**2. Experiment procedure and method**
2.1 Experiment

Two sandboxes (Fig. 2a and b) packed by translucent silica medium are prepared in

laboratory to test different criteria of REV quantification. PCE is selected as a typical
DNAPL contaminant used in experiments. 2D sandbox is composed by three aluminum
interior frames and two glass walls, which thickness is 1.6cm. The dimensions of
sandboxes used in Experiments-I and II are 20 (width) ×15 (height) and 60 (width)×45
(height)F40/50 and F20/30 mesh translucent silica sands are used for background material
for Experiments-I and II. To make the translucent silica fully saturated by water in a flow
field close to natural groundwater environment (Erning et al., 2012), water flow at flow
velocity of 0.5 m/d is set from left to right in laboratory sandbox experiments (Fig. 2a and
b). Water is restricted in a sandbox that the top and bottom layers of sandbox are packed
by F70/80 mesh translucent silica as capillary barriers. Light source is placed behind the
sandbox to make light penetrate through translucent media and capture emergent light
intensity using a thermoelectrically air-cooled charge-coupled device (CCD) camera (Fig.
1a). Afterward, PCE is injected into sandboxes from the injection needle at constant rate
of 0.5 mL /min for two experiments. Detailed experimental conditions are listed in Table1.



2.2 Light transmission techniques
By means of light transmission technique (Fig. 1a), DNAPL and water saturation can
be obtained rapidly and in real-time, which greatly help to explore mechanism of
groundwater flow and contaminant migration in porous media. When light passes through
translucent materials of a given thickness, the emergent light intensity after the absorptive
and interfacial losses can be expressed as (Niemet and Selker, 2001; Bob et al., 2008; Wu
et al., 2017):

$$I_T = CI_0 ( \prod \tau_j ) exp( -\sum \alpha_i d_i ) \tag{1}$$

where $I_0$ is the original light intensity; $C$ is a constant of correction for light emission
and light observation; $\tau_j$ is the transmittance when light penetrate from phase $i$ to $i+1$;
$\alpha_i$ is the absorption coefficient when light penetrate in phase $i$; $d_i$ is the length of light
penetration path in phase $i$.
To derive the porosity, the 2D translucent porous medium should be only saturated
by water. Consequently, the emergent light intensity can be expressed as:

$$I_s = CI_0 \tau_{s,w}^{2k_o} \exp(-\alpha_s k_s d_s) \tag{2}$$

where $\tau_{s,w}^{2k_o}$ is the transmittance of solid-water interface; $\alpha_s$ is solid particles
absorption coefficient; $d_s$ is median diameter of the solid particles; $k_o$ is the number of
pores across light penetration path; $k_s$ is the number of solid particles across light
penetration path.
If we arbitrarily select an infinitesimal element, its area $A_o$ approaches zero
$(A_o \rightarrow 0)$ from the 2D translucent porous media (Fig. 1b), and suppose the infinitesimal
element with thickness $L_T$ containing solid particles and pores that can be regarded as





lamellar structure (Fig. 1c), we can obtain the following relationships:
$$\theta A_o L_T = A_o k_o d_o \tag{3}$$

$$k_s d_s + k_o d_o = L_T \tag{4}$$

where $d_o$ is the median diameter of pores; $\theta$ is porosity.

Substituting Eq. (3) and Eq. (4) into Eq. (2), the relationship between emergent

light intensity and porosity can be achieved (Wu et al., 2017):
$$lnI_s = \beta + n\gamma \tag{5}$$

where   $\beta = \ln(\dfrac{CI_s}{e^{\alpha_s d_s L_T}})$   and   $\gamma = \ln(\tau_{s,w}^{\frac{2L_s}{d_o}} e^{\alpha_s L_T})$ .   $\beta$  and  $\gamma$  can  be  determined  from
experimental data, then porosity can be obtained.

The density and tortuosity are derived as (Wu et al., 2018):

$$\rho = \theta \rho_w + (1.0 - \theta)\rho_s \tag{6}$$

$$\tau = 1 + \frac{\pi - 2}{\sqrt{\dfrac{\pi}{1-\theta}}} \tag{7}$$

where $\rho$ is the density of translucent porous media; $\rho_w$ is the density of water; $\rho_s$ is the
density of solid particles; $\tau$ is tortuosity .

The saturation of dense nonaqueous phase liquid (DNAPL) was quantified by light

transmission technique based on light pass through translucent materials (Niemet and
Selker, 2001; Bob et al., 2008):
$$S_o = \frac{\ln I_s - \ln I_T}{\ln I_s - \ln I_{oil}} \tag{8}$$

where $S_o$ is the saturation of DNAPL; $I_s$ is the light intensity after light penetration
through translucent porous when all pores are fully saturated by water; $I_{oil}$ is the light
intensity when all pores are fully saturated by DNAPL; $I_T$ is the light intensity after


penetration through translucent materials. After quantification of PCE saturation,
PCE-water interfacial area ($A_{OW}$) can be obtained based on the method proposed by Wu et
al. (2017), where the unit of $A_{OW}$ is $cm^{-1}$.
2.3 Criteria of REV quantification
The REV is defined as the volume range in which all material characteristics are
factored into the average and associated values approach single and constant (Brown
and Hsies, 2000). In the range of REV, the value of one associated property will meet
the condition:
$$\frac{\partial Y(L_i)}{\partial L}\Big|_{L_i=L_o} = 0 \tag{9}$$

where the $Y(L_i)$ is the value of an associated property when system scale is $L_i$; $L_i$ is the
value of system scale; $L_o$ is the scale range of REV, $L_{min} < L_o < L_{max}$; $L_{max}$ is upper
boundary of REV; $L_{min}$ is lower boundary of REV scale. According to the Eq. (9),
when the measured scale ($L_i$) reaches REV range, the derivative $\frac{\partial Y(L_i)}{\partial L} \to 0$ will tend
to zero. As a matter of fact, most previously used criteria were applied to estimate REV
based on this requirement. The REV estimation criteria tested in this study are
illustrated in Table 2.
To evaluate the REV of porosity, the coefficient of variation ($C_V^i$) (Table 2) is
utilized to estimate the variability (Nordahl and Ringrose, 2008):
$$C_V^i = \frac{\hat{s}_i}{\overline{\varphi}_i} \tag{10}$$

where $i$ is the cuboid window (Fig. 1b) increment number; $\varphi$ is the measured variable,
such as porosity; $\hat{s}_i$ is the standard deviation of sub-grids' variable in different





measured volume or scale; $\overline{\varphi}_i$ is the arithmetic average of the variable values in the
sub-grids. When number of sub-grids ($N$) is less than 10, a correction is utilized to
replace Eq. (10). According to Nordahl and Ringrose (2008), $0 < C_V^i < 0.5$ is defined
as homogeneous and $C_V^i = 0.5$ can be used as criterion to identify the REV scale.
Similarly, for porosity of translucent silica, entropy dimension ($DI^i$) (Table 2) is
utilized for REV analysis and estimation (Martínez et al., 2007), which is defined as:
$$DI^i \approx \frac{\sum_{j=1}^{m(i)} \mu_j(L_\varepsilon) \log \mu_j(L_\varepsilon)}{\log L_\varepsilon} \qquad (11)$$

where, $L_\varepsilon$ is the scale of sub-grid; "≈" indicates the asymptotic equivalence as $L_\varepsilon \to 0$
(Martínez et al., 2007); $j$ is the ordinal number of sub-grid in measured cuboid window
(Fig. 1b) of increment number $i$; $m(i)$ is the total number of sub-grids in measured
cuboid window (Fig. 1b) of increment number $i$; $\mu_j(\varepsilon)$ is the proportion of the
variable of sub-grid $j$ in the whole measured cuboid window $i$. The right hand side of Eq.
(11) is the simplification of Shannon entropy of all sub-grids, in which $DI^i$ can be
considered as the average of logarithmic values of the variable distribution weighted by
$\mu_j(L_\varepsilon)$ to quantify the degree of medium heterogeneity. Using Eq. (11), a series of values
of $DI^i$ ($i$=1,2,3…) are obtained for each measured cuboid window (Fig. 1b) of
increment number $i$. For estimation of the REV in a porous medium, the relative
increment of entropy dimension and associated criterion of REV identification are
respectively expressed as:
$$RI^i = \left| \frac{DI^j - DI^{j-1}}{DI^{j-1}} \right| \times 100 \qquad (12)$$

$$RI^i \leqslant 0.2 CV_{DI} \qquad (13)$$

where $CV_{DI}$ is the coefficient of variation of $DI^i$ series ($i$=1,2,3…), which is



calculated through $CV_{DI} = (\hat{s}_{DI} / \overline{DI}) \times 100$; $\overline{DI}$ is the mean value of the $DI^i$ series;
$\hat{s}_{DI}$ is the standard deviation of the $DI^i$ series.

To achieve the REV for multiple system variables, such as porosity, moisture

saturation and air-water interfacial areas in an unsaturated porous medium, a criterion
named the relative gradient error (Table 2) was applied (Costanza-Robinson et al., 2011):
$$\varepsilon_g^i = |\frac{\varphi^{i+1} - \varphi^{i-1}}{\varphi^{i+1} + \varphi^{i-1}}| \frac{1}{\Delta L} \tag{14}$$

where $\varepsilon_g^i$ is relative gradient error; $\Delta L$ is the measured cuboid window size increment
length for REV estimation. Usually, $\varepsilon_g^i$ less than 0.2 (Costanza-Robinson et al., 2011) is
utilized to identify a REV sizes.

A new criterion based on the required condition of REV is proposed to estimate

the REV range for the translucent silica in this study:
$$\chi^i = \frac{|\delta_{i+1} - \delta_{i-1}|}{\delta_i \Delta L} \tag{15}$$

where $\delta^i$ is the dimensionless range, $\delta^i = \frac{\phi(L_i)_{max} - \phi(L_i)_{min}}{\overline{\phi(L_i)}}$; $\varphi(L_i)_{max}$ is the
maximum value of the variable on the volume scale $L_i$; $\varphi(L_i)_{min}$ is the minimum value
of the variable on the volume scale $L_i$; $\overline{\varphi(L_i)}$ is the mean value of the variable on the
volume scale $L_i$. Brown and Hsieh (2000) suggested $\delta^i = \frac{\phi(L_i)_{max} - \phi(L_i)_{min}}{\overline{\phi(L_i)}} \ll 1$ can
be used for REV estimation. In fact, the calculated value of $\delta^i$ mostly is less than 1,
while $\delta^i \ll 1$ is hard to be used to identify the REV scale for realistic materials, such
as the translucent silica used in this study. The value limit of $\chi^i$ used for REV estimation
also is explored in this study.



226   In this study, criteria for the coefficient of variation ($C_V^i$), entropy dimension

227 ($DI^i$), the relative gradient error ($\varepsilon_g^i$) and the new criterion ($\chi^i$) are all applied in REV

228 estimation for porosity and PCE saturation. Corresponding REV plateau identification

229 effects are compared to select the best criterion for REV quantification.

230  **3. Results and discussion**

231 3.1 REV identification effect of different criteria

232 *3.1.1 The coefficient of variation*

233   Emergent light intensity distributions of translucent silica for two experiments, which

234 had been fully saturated by water, was obtained by a thermoelectrically air-cooled

235 charge-coupled device (CCD) camera (Niemet and Selker, 2001; Bob et al., 2008). The

236 porosity, density, tortuosity and PCE saturation for two experiments are derived by light

237 transmission technique as shown in Figs. 3a and b. The PCE spreads from the injecting

238 point shaped like a drop of water at t=1.44 min (Fig. 3b). In 2D sandboxes for two

239 experiments, PCE plume infiltrates in translucent silica sands infiltration paths and PCE

240 plumes reach the bottom after t=80 min.

241   Every pixel with small scale could be approximated as infinitesimal element in high

242 resolution image to apply light transmission techniques. As consequence, porosity of

243 translucent silica was derived with light transmission technique through Eq. (5) (Fig. 2c).

244 The whole 2D translucent silica area was numerically discretized that every cell had the

245 uniform dimensions of 0.015m×0.015m. The cuboid window (Fig. 1d) was utilized to

246 quantify the variables (porosity, density, tortuosity, PCE saturation, PCE-water interfacial

247 area) of every cell as measured scale was increased. In detail, the measured cuboid





window scale was increased from the center of each cell and associated value of variable
was calculated from the high resolution porosity of 2D translucent silica derived by light
transmission technique. Observation cells were selected from the discretized cells (Fig. 3b)
of which the cells I-1~2 and II-1~2 belong to Experiments-I and II, respectively. Porosity
and PCE saturation variation curves of these observation cells with increasing measured
cuboid window scale were shown in Fig. 4a and b. However, for all observation cells from
translucent silica, the REV plateaus were not obvious to be objectively judged visually,
which made REV plateaus hard to identify effectively by original variation curves for
porosity and PCE saturation (Figs. 4a and b).

To make the REV plateau more explicit, different criteria of REV quantification

are utilized. The coefficient of variation ($C_V^i$) of porosity and PCE saturation fluctuating
with increase of measured cuboid window size is shown in Fig. 4. The measured cuboid
window scale is limited to the dimensions of cells in discretization of 2D translucent
silica. The observation cells show various characteristics of variation tendency for the
coefficient of variation ($C_V^i$). The θ and S$_o$ variation curves of coefficient of variation ($C_V^i$)
for all observation cells do not reach stable values as those shown in Figs. 4a and b, the
beginning of the REV flat plateau is not easy to identify, the coefficient of variation ($C_V^i$)
is not suitable for REV estimation. According to the heterogeneity definition by Corbett
and Jensen (1992), the heterogeneity of materials is defined by $C_V^i$ magnitude that
$0 < C_V^i < 0.5$ is classed as homogeneous medium, $0.5 < C_V^i < 1.0$ is classed as
heterogeneous medium and $1.0 < C_V^i$ is classed as strong heterogeneous medium. For
the coefficient of variation ($C_V^i$) magnitude in Figs. 4a-f, the $C_V^i$ values are all below





0.5 that the criterion $C_V^i = 0.5$ is unable to identify the REV scale for translucent
silica.

*3.1.2 Entropy dimension*

Entropy dimension ($DI^i$) is utilized by Martínez et al. (2007) for multifractal
analysis of a porous medium porosity and REV estimation. In this study, entropy
dimension ($DI^i$) is tested to avoid unclear REV plateau in porosity curves. The entropy
dimension ($DI^i$) of porosity is calculated by Eq. (11). Variation curves of entropy
dimension ($DI^i$) for all observation cells (Fig. 2a) are presented in Fig. 4. The curves of
entropy dimension ($DI^i$) of porosity and PCE saturation generally result in the increasing
trend curves which makes REV estimates become very difficult and invalid. Entropy
dimension ($DI^i$) was quickly increased with increasing of measured cuboid window size.
Compared to the coefficient of variation ($C_V^i$) of porosity and PCE saturation, entropy
dimension ($DI^i$) increased step by step without opposite fluctuation tendency in the
variation curves as length scale of measured cuboid window increased simultaneously. In
general, REV plateau in region II (Fig. 1c) of porosity is not obvious for the entropy
dimension ($DI^i$) curves of all observation cells from two experiments, which suggests
REV scales is uneasy to identify for translucent silica using entropy dimension ($DI^i$) by
light transmission technique.

*3.1.3 The relative gradient error*

The relative gradient error ($\varepsilon_g^i$) of porosity and PCE saturation is calculated by Eq.
(14). The variation of $\varepsilon_g^i$ at different measured cuboid window scales are shown in Fig.





4 for all observation cells in the 2D translucent silica. For all $\varepsilon_g^i$ curves at observation
cells from experiments, the REV plateaus in region II (Fig. 1a) are more clear than the
variation curves based on the criteria of $C_V^i$ and $DI^i$. Apparently, erratic variations of
the relative gradient error ($\varepsilon_g^i$) at small measured cuboid window scales are observed
for all $\varepsilon_g^i$ curves as the characteristic of REV region I in Fig. 1c. When measured
cuboid window scale further increases for all observation cells, the variability and
magnitude of the relative gradient error ($\varepsilon_g^i$) decrease well and factored into average,
which can be identified as REV plateau in region II (Fig. 1c). The relative gradient error
($\varepsilon_g^i$) makes the REV plateau quantification convenient for all observation cells. At the
measured cuboid window size above the REV plateau, $\varepsilon_g^i$ curves result in large
variability for observation cells I-1~2. These findings suggest that that the relative
gradient error ($\varepsilon_g^i$) can make the REV plateau more obvious, which greatly contribute
to convenient and applicable REV quantification for translucent silica by light
transmission technique. However, random fluctuations exist in $\varepsilon_g^i$ curves visually,
which make the REV plateau uneasy to identify accurately.
*3.1.4 The new criterion ($\chi^i$)*

$\chi^i$ of porosity and PCE saturation changing with measured cuboid window size is

shown in Fig. 4. Like the region I (Fig. 1c), erratic and random fluctuations appears at
small measured cuboid window sizes and $\chi^i$ increases with the increase of the measured
cuboid window size. When measured scale increases, the values of $\chi^i$ for all observation
cells appear fast reduction and rapidly tend to steady, which exhibit the characteristic of
REV plateau as measured scale reaches region II. The $\chi^i$ for observation cells restore the



erratic variation state of increasing trend after measured cuboid window size exceeding the
REV plateau. As shown in the variation curves of $\chi^i$ for all observation cells, the beginning
of the REV flat plateaus can be identified easily, suggesting $\chi^i$ is more convenient and
reliable than other methods for REV estimation. All observation cells show similar
variation curves of $\chi^i$ that low value intervals are quite apparent, indicating that $\chi^i$ is very
effective to make the REV plateau obvious for translucent silica used in this study. Using
the criterion of $\chi^i$, the REV plateau of region II is flat, which is easily identified,
compared with other criteria for observation cells (Figs. 4a and b).
3.2 REVs of material properties

Based on the REV plateau identifications using the coefficient of variation

($C_V^i$), entropy dimension ($DI^i$), the relative gradient error ($\varepsilon_g^i$) and the proposed new
criterion $\chi^i$ in Figs. 4a and b, the new criterion $\chi^i$ appears to be the most appropriate
criterion for REV plateau identification. Even though the relative gradient error ($\varepsilon_g^i$) can
also make REV plateau obvious, but various random fluctuations weaken the method and
reduce the associated accuracy. Therefore, REVs of porosity, density, tortuosity and PCE
plume are estimated using the new criterion $\chi^i$ in the following study.

In fact, large number of discretized cells in the 2D translucent silica for two

experiments are quantified using the new criterion $\chi^i$, which is convenient to examine the
regularities for REV sizes and related factors. Using the new criterion $\chi^i$, the REV
estimation is conducted based on Eq. (15). Fig. 5a shows minimum REV sizes of porosity,
density and tortuosity quantified by $\chi^i$ for all cells of two experiments. Associated
statistical analysis for REVs is illustrated in Fig. 5b, where circular points represent





frequency and triangular points represent cumulative frequency. Frequency of REVs is
dense in the middle and sparse on both sides, so the distribution of REVs can be fitted by
Gaussian equation:
$$F = \omega + \frac{1}{\sqrt{2\pi}\delta} e^{-\frac{(\text{REV}-\nu)}{2\delta^2}}$$
(16)

where F is the frequency of REV; $\omega$, $\delta$ and $\nu$ are fitted parameters of the model.
After regression analysis, the derived models for REV frequency are listed in Table 3.
The coefficients of determination ($R^2$) of models for REVs of porosity and density all
exceed 0.93. $R^2$ for REV of tortuosity for two experiments exceed 0.7. Moreover, the
computed cumulative frequency based on models fit cumulative frequency from
experimental results well in Fig. 5b.
The minimum REV size frequency for porosity appears a peak between 4.0 mm and
5.0 mm for Experiment-I. As minimum REV size of porosity increases, corresponding
frequency continuously decreases. Further, smooth convex shape of cumulative frequency
is observed for minimum REV size of porosity (Fig. 5b). Most minimum REV sizes of
translucent silica distributed in 0.0-7.0mm. For density of translucent silica sand,
associated REV frequency appear high values between 2.0~3.0 mm. For the REV sizes of
tortuosity, minimum REV sizes distribute in 0.0~6.0 mm. Compared with Experiment-I
(F40/50 mesh translucent silica sand), the frequency of REV for Experiment-II (F20/30
mesh translucent silica sand with larger porosity) show flat shape and has larger value of
standard deviation, especially for REV of porosity. Fig. 5b shows that translucent silica
with larger porosity will achieve border distribution of minimum REV sizes distribution
compared to translucent silica with relative lower porosity. The mean REV sizes of


porosity, density and tortuosity for Experiment-I are 4.35 mm, 2.89 mm and 3.65 mm,
respectively. All mean REV sizes of these variables for Experiment-II are larger than
REVs of Experiments-II, which corresponding mean REV sizes are 8.05 mm, 2.97 mm
and 4.30 mm. These results suggest translucent porous media with higher porosity lead to
larger values of mean and standard deviation for REV sizes.
3.3 REVs of $S_o$ and $A_{OW}$ for PCE plume
The minimum REV sizes of $S_o$ and $A_{OW}$ obtained using new criterion $\chi^i$ are shown in
Figs. 6a and b. To analyze the regularity of REV distribution for PCE plume, the mass
center coordinate of PCE plume for two experiments are quantified for Experiments-I and
II. The mass center coordinate are calculated as:
$$x_m = \frac{M_{10}}{M_{00}} \qquad (17)$$

$$Z_m = \frac{M_{01}}{M_{00}} \qquad (18)$$

where $x_m$ is x coordinate of mass center for PCE plume; $z_m$ is z coordinate of mass center
for PCE plume; $M_{00}$, $M_{10}$ and $M_{01}$ are computed using definition of spatial moment ($M_{ij}$),
$M_{ij} = \int_{x_0}^{x_1} \int_{z_0}^{z_1} \theta(x,z) S_o(x,z,t) x^i z^j d_x d_z$ ; $x_0$ and $z_0$ are minimum limits of x axis and z axis; $x_1$
and $z_1$ are maximum limits of x axis and z axis; $\theta(x,z)$ is the porosity at point $(x,z)$; $S_o(x,z,t)$
is PCE saturation of point $(x, z)$ at time t.
The mass center coordinate of PCE plume derived by Eq. (18) is shown in Fig. 7a.
Afterward, the average value of REVs ($\overline{REV}$) and associated distance ($d_m$) from mass
center to corresponding cells contained in PCE plume at t=1523 min are presented in Fig.
7a. Regression analysis is performed for average REVs of PCE plume and $d_m$, where fitted



models and associated $R^2$ for Experiments-I and II are listed in Table 4. Simultaneously,
the fitted equations between $\overline{REV}$ and $d_I$ (the distance from injection point to cell
contained in PCE plume) also are derived by regression analysis. From the results in Fig.
7a, $\overline{REV}$ of $S_o$ and $A_{ow}$ appear a peak and then decrease with increasing of $d_m$ and $d_I$ for
Experiment-I. $\overline{REV}$ of $S_o$ and $A_{ow}$ for Experiment-I all firstly increase and then decrease
with the increasing of $d_m$ and $d_I$, while $\overline{REV}$ of PCE plume presents apparent decreasing
tendency as $d_m$ and $d_I$ increase for Experiment-II. In addition, the value of $A_{OW}$-REV
mostly is higher than the value of $S_o$-REV for two experiments.
The mean and standard deviation of REVs of PCE plume during 0~1523 min derived
by statistical analysis (Fig. 7b). Compared with REVs of PCE plume for Experiment-I,
Experiment-II (F20/30 mesh translucent silica sand with higher porosity) has larger value
of mean and standard deviation of REVs. Besides, the relationship between REVs and
PCE saturation are fitted by regression analysis, where fitted equation and $R^2$ for two
experiments are listed in Table 5 and Fig. 7b. With increasing of PCE saturation, REV of
$S_o$ appear decline trend for two experiments. However, REV of $A_{ow}$ increases when $S_o$
increases for both two experiments (Fig. 7b). On the other hand, REV of $S_o$ for
Experiment-II is higher than corresponding REV for Experiment-I, while Experiments-I
and II have similar values of $A_{OW}$-REV (Fig. 7b). These results suggest higher porosity
will lead to high value of $S_o$-REV and the relationship between REVs of PCE plume and
dm, $d_I$.
**4. Conclusions**
In this study, a new criterion $\chi^i$ is proposed to identify the REVs of translucent porous





media and inner contaminant transformation based on previous criteria. The REV plateaus
of observation cells selected from two experiments of PCE transport are hard to judge
visually from the porosity and PCE saturation curves. From the REV identification effects
of different criteria, the REV flat plateau is difficult to identify by coefficient of variation
($C_V^i$) and entropy dimension ($DI^i$), indicting the coefficient of variation ($C_V^i$) and entropy
dimension ($DI^i$) are not suitable for REV estimation of translucent porous media. The
relative gradient error ($\varepsilon_g^i$) can make REV plateaus of all kinds of translucent silica
explicit in variation curves, but random fluctuations weaken REV plateau identification. In
comparison with these previous criteria, the beginning and ending of the REV flat plateaus
could be easily and directly identified in the curves based on the new criterion $\chi^i$,
suggesting the new criterion $\chi^i$ is more convenient and effective for REV estimation. In
this study, REVs of porosity, density, tortuosity, and PCE plume are estimated using the
new criterion $\chi^i$.

Statistical results of minimum REV scales quantified by new criterion $\chi^i$ reveal

cumulative frequencies of porosity, density and tortuosity all have smooth convex shapes.
Models based on Gaussian equation are built for the distribution of REVs of porosity,
density and tortuosity, which porous media with larger porosity leads to larger values of
mean and standard deviation for REV sizes of media properties. For REVs of PCE plume,
result suggested larger porosity lead to larger value of mean and standard deviation.
Regression analysis is performed to study the regularity for distribution of REVs, where
fitted relationship between REVs and $d_m$, $d_l$ are derived for PCE plume. $\overline{REV}$ of $S_o$ and
$A_{ow}$ firstly increase and then decrease with the increasing of $d_m$ and $d_l$ for Experiment-I



whose sandbox packed by translucent porous media with relatively lower porosity.
However, $\overline{\text{REV}}$ of $S_o$ and $A_{ow}$ directly decrease with the increment of $d_m$ and $d_I$ when
porosity became larger for Experiment-II. Significantly, REV size of $S_o$ presented
decreasing trend as $S_o$ increases, while increasing tendency appeared for REV size of $A_{ow}$.
Through regression analysis, the fitted equations between REVs of PCE plume and PCE
saturation are derived for two experiments. Implications of these finding are essential for
quantitative investigation of scale effect of porous media and contaminant transformation.
The fluid migration and transform in porous media can be simulated accurately according
to the REV estimation results using light transmission technique and the appropriate
criterion $\chi^i$.

**Code and data availability**

The codes and data for this paper are available by contacting the corresponding author at
jfwu@nju.edu.cn.

**Author contributions**

Ming Wu: Conceptualization, Methodology, Writing;
Jianfeng Wu: Conceptualization, Methodology, Writing;
Jichun Wu: Conceptualization;
Bill X. Hu: Conceptualization, Writing.



**Declaration of interests**

The authors declare that they have no known competing financial interests or personal
relationships that could have appeared to influence the work reported in this paper.

**Acknowledgments**

We acknowledge support by the National Key Research and Development Plan of

China (2016YFC0402800), the National Natural Science Foundation of China (41902246,
41730856 and 41772254), the National Natural Science Foundation of China-Xianjiang
Project (U1503282) and the China Postdoctoral Science Foundation (2017M622905).

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





**Table 1.** Experimental conditions

| Experiment | I | II |
|---|---|---|
| Sandbox dimensions (cm) | 20×15 | 60×45 |
| Packed translucent silica sand | F40/50 | F20/30 |
| Median grain diameter (mm) | 0.36 | 0.72 |
| Permeability (m$^2$) | $4.25\times10^{-11}$ | $1.35\times10^{-10}$ |
| $V_{PCE}$ (ml) | 9 | 32 |
| Injection rate (ml/min) | 0.5 | 0.5 |







**Table 2.** Criteria of REV estimation

| Criterion | Equation |
| --- | --- |
| The coefficient of variation | $C_V^i = \dfrac{\hat{s}}{\overline{\varphi}_i}$ |
| entropy dimension | $DI^i \approx \dfrac{\sum_{j=1}^{m(i)} \mu_j(L_\varepsilon)\log\mu_j(L_\varepsilon)}{\log L_\varepsilon}$ |
| the relative gradient error | $\varepsilon_g^i = \left\| \dfrac{\varphi^{i+1} - \varphi^{i-1}}{\varphi^{i+1} + \varphi^{i-1}} \right\| \dfrac{1}{\Delta L}$ |
| New criterion | $\chi^i = \dfrac{\|\delta_{i+1} - \delta_{i-1}\|}{\delta_i \Delta L}$ |







**Table 3.** The fitted equations of frequency for REVs of porosity, density and tortuosity

| Experiment | I | II |
|---|---|---|
| θ-REV | $F = -2.01 \times 10^{-12} + \dfrac{1}{\sqrt{2\pi} \times 1.50} e^{-\frac{(REV-4.35)^2}{2*1.50^2}}$ | $F = -5.30 \times 10^{-3} + \dfrac{1}{\sqrt{2\pi} \times 3.35} e^{-\frac{(REV-8.05)^2}{2*3.35^2}}$ |
| | ($R^2$=0.955) | ($R^2$=0.932) |
| ρ-REV | $F = -7.51 \times 10^{-26} + \dfrac{1}{\sqrt{2\pi} \times 1.14} e^{-\frac{(REV-2.89)^2}{2*1.14^2}}$ | $F = -3.18 \times 10^{-12} + \dfrac{1}{\sqrt{2\pi} \times 1.71} e^{-\frac{(REV-2.97)^2}{2*1.71^2}}$ |
| | ($R^2$=0.969) | ($R^2$=0.989) |
| τ-REV | $F = -2.76 \times 10^{-15} + \dfrac{1}{\sqrt{2\pi} \times 1.42} e^{-\frac{(REV-3.65)^2}{2*1.42^2}}$ | $F = -8.55 \times 10^{-8} + \dfrac{1}{\sqrt{2\pi} \times 2.15} e^{-\frac{(REV-4.30)^2}{2*2.15^2}}$ |
| | ($R^2$=0.774) | ($R^2$=0.850) |

[*]F represents the frequency of REV, θ represents porosity, ρ represents density, τ
represents tortuosity






**Table 4.** The fitted equations between average value of REV and $d_I$, $d_m$

| Experiment | I | | II | |
|---|---|---|---|---|
| | $S_o$-REV | $A_{OW}$-REV | $S_o$-REV | $A_{OW}$-REV |
| $d_m$ | $\overline{REV}=-1.67\times10^{-3}d_m^2$ $+0.193d_m+2.72$ ($R^2$=0.807) | $\overline{REV}=-6.10\times10^{-4}d_m^2$ $+5.82\times10^{-2}d_m+7.20$ ($R^2$=0.401) | $\overline{REV}=-4.08\times10^{-5}d_m^2$ $+1.50\times10^{-2}d_m+7.54$ ($R^2$=0.655) | $\overline{REV}=-1.92\times10^{-5}d_m^2$ $+4.47\times10^{-3}d_m+9.46$ ($R^2$=0.616) |
| $d_I$ | $\overline{REV}=-1.97\times10^{-3}d_I^2$ $+0.245d_I+1.12$ ($R^2$=0.832) | $\overline{REV}=-1.47\times10^{-3}d_I^2$ $+0.205d_I+1.84$ ($R^2$=0.733) | $\overline{REV}=-3.94\times10^{-5}d_I^2$ $+7.80\times10^{-3}d_I+8.50$ ($R^2$=0.327) | $\overline{REV}=-1.92\times10^{-5}d_m^2$ $+4.47\times10^{-3}d_m+9.46$ ($R^2$=0.616) |

* $\overline{REV}$ is the average value of REV size, $d_m$ is the distance from mass center of PCE plume
to the cell contained in PCE plume, $d_I$ is the distance from injection point to the cell
contained in PCE plume






**Table 5.** The fitted relationship between REV and $S_o$

| Experiment | I | II |
|---|---|---|
| $S_o$-REV | $REV = -2.13 \times \ln S_o + 0.876$ | $REV = -0.961 \times \ln S_o + 1.09$ |
| | ($R^2$=0.466) | ($R^2$=0.415) |
| $A_{OW}$-REV | $REV = 2.27e^{2.70*S_o}$ | $REV = 1.70e^{3.30*S_o}$ |
| | ($R^2$=0.366) | ($R^2$=0.500) |




**Figure Captions**

**Figure 1.** (a) System Device for acquisition of properties of translucent material; (b) The
infinitesimal selected from translucent porous media packed in 2D sandbox; (c)
Variable changes as measured scale (L) increment in conceptual curve
(Costanza-Robinson et al., 2011); (d) Scale effect and the cuboid image window
geometry.
**Figure 2.** (a) The system sandbox equipment of Experiment-I; (b) The system sandbox
equipment of Experiment-II
**Figure 3.** (a) The emergent light intensity, porosity, permeability and tortuosity of 2D
translucent silica sand for Experiments-I and II; (b) The PCE saturation of
Experiments-I and II during 0~1523 min and observation cells
**Figure 4.** (a) The change of porosity ($\theta$), associated coefficient of variation ($C_V^i$), entropy
dimension ($DI^i$), the relative gradient error ($\varepsilon_g^i$), and new criterion-$\chi^i$ for
observation cells as cuboid window scale (L) increases; (b) The change of PCE
saturation ($S_o$), associated $C_V^i$, $DI^i$, $\varepsilon_g^i$, and $\chi^i$ for observation cells as cuboid
window scale (L) increases
**Figure 5.** (a) The distributions of minimum REV sizes of porosity, sand density and
tortuosity for Experiments-I and II; (b) The frequency of minimum REV sizes of
Experiments and fitted models
**Figure 6.** (a) The distributions of $S_o$-REV sizes during 0~1523 min for Experiments-I and
II; (b) The distributions of AOW-REV sizes during 0~1523 min for Experiments-I
and II





601 **Figure 7.** (a) The mass center coordinate of PCE plume and the change of average REV

602   size as the distance $d_l$, $d_m$ increases; (b) The mean, standard deviation of $S_o$-REV

603   and $A_{OW}$-REV during 0~1523 min and fitted relationship between REV sizes and

604   $S_o$ for Experiments-I and II






**Fig. 1**

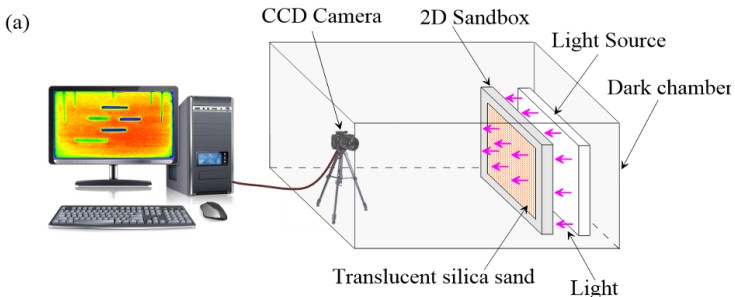

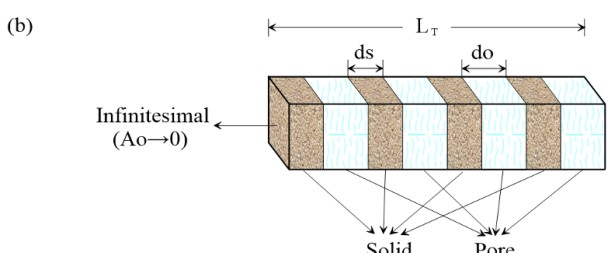

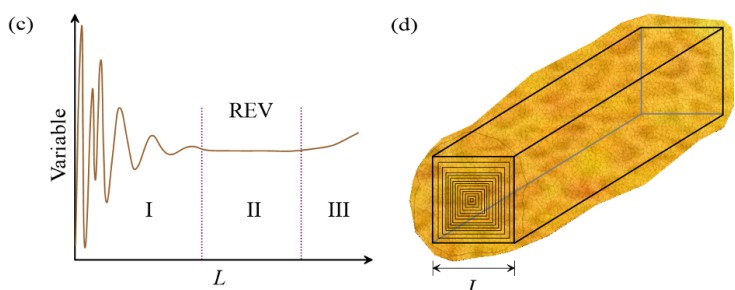




**Fig. 2**

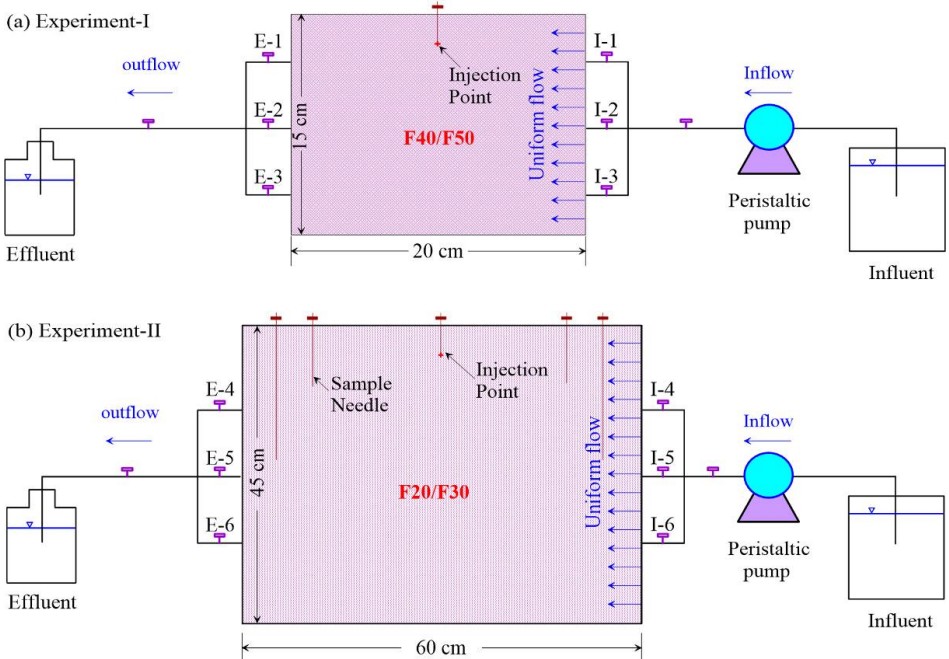





**Fig. 3**

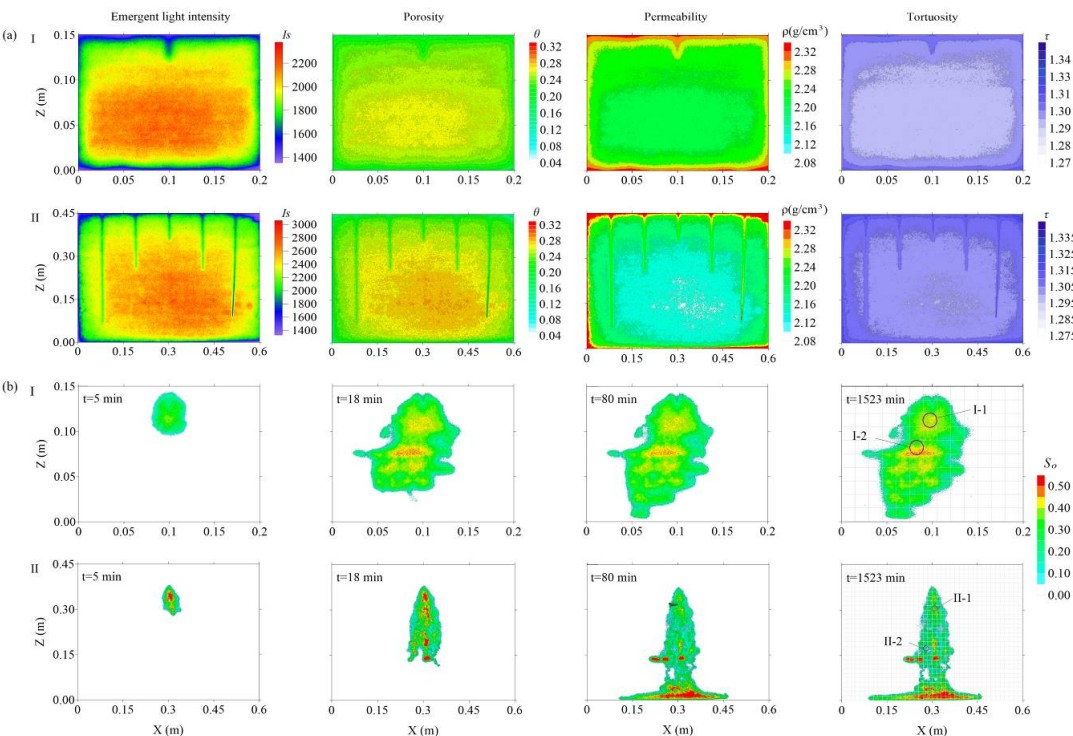







**Fig. 4**

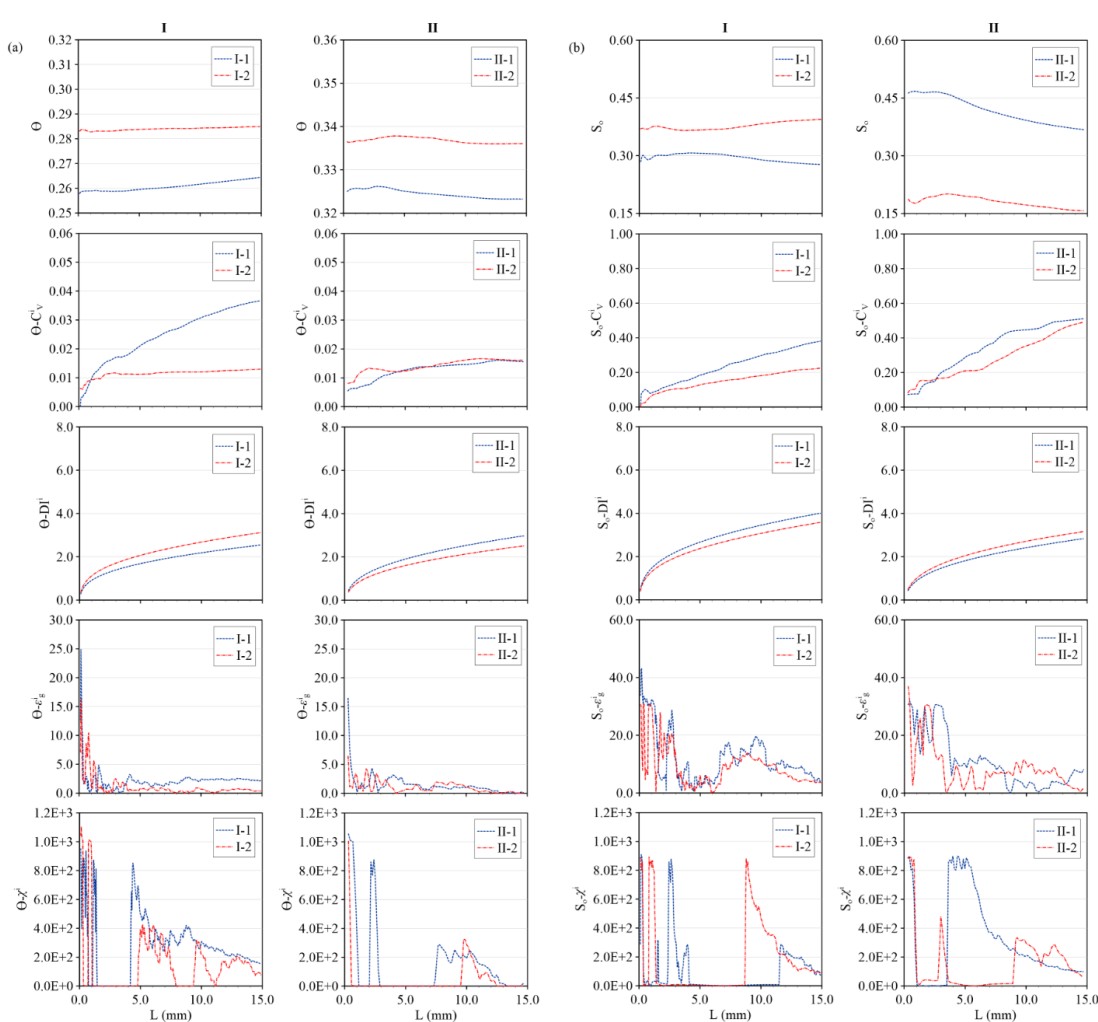








**Fig. 5**

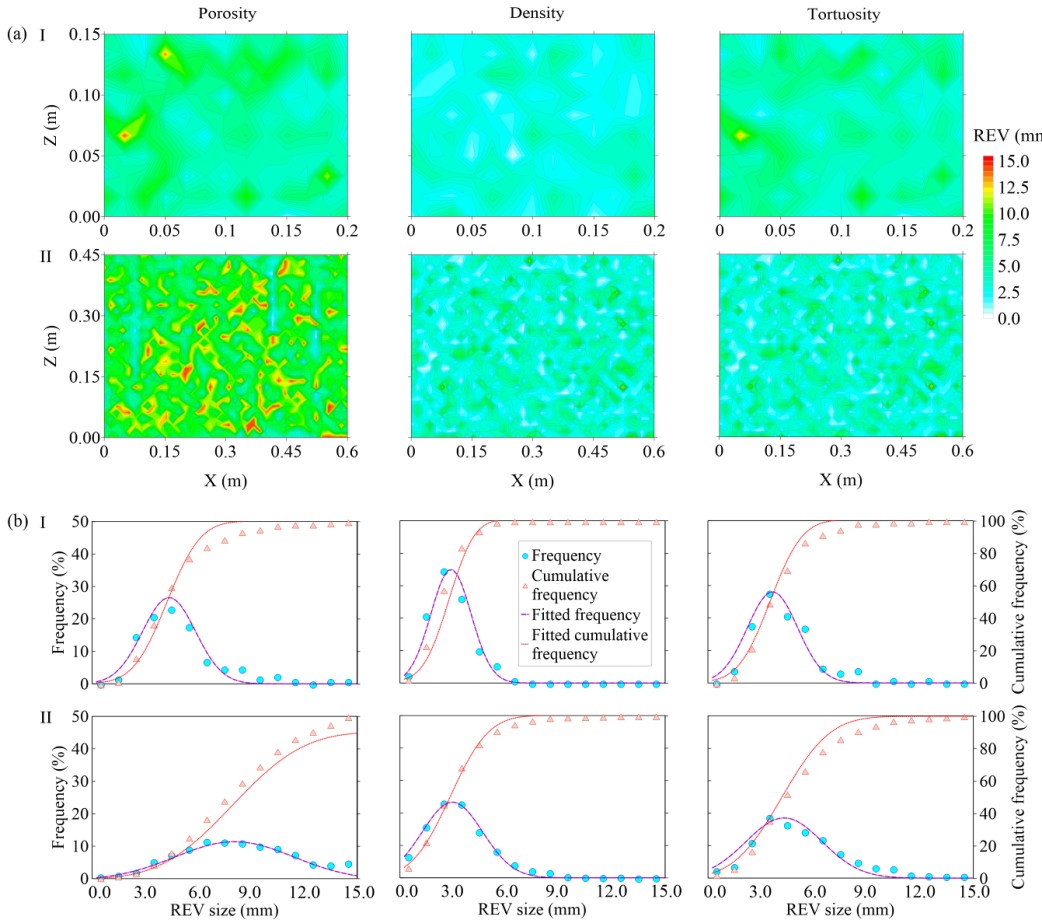

**Fig. 6**

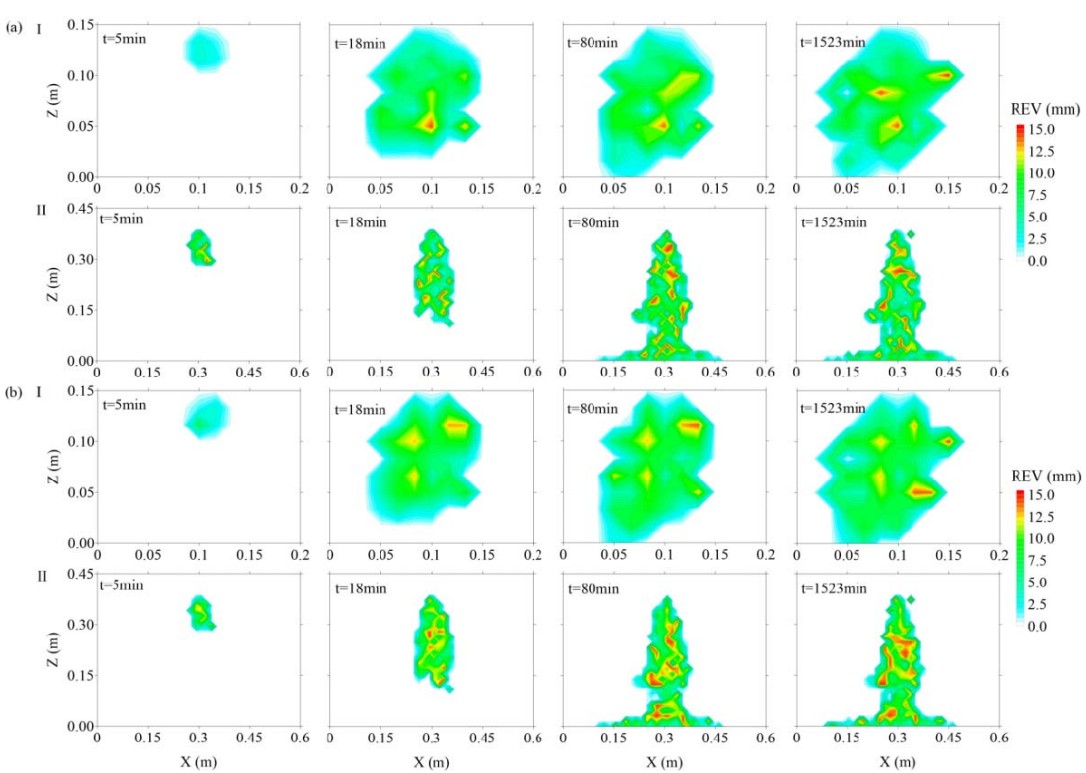






**Fig. 7**

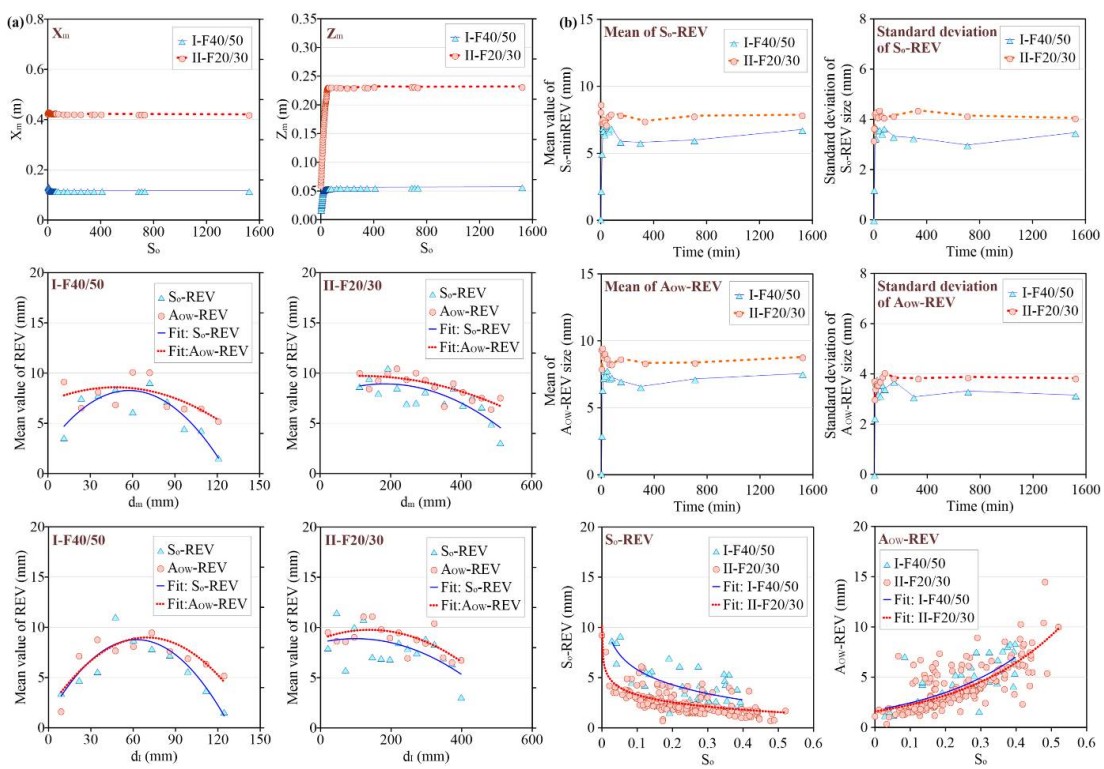
