# Peer review of "A new criterion for determining the representative elementary volume of translucent porous media and inner contaminant"

_Hydrology and Earth System Sciences, 2020_

## Referee Comment (RC1) · Anonymous Referee #1 · 24 Jun 2020

This paper presents a study of determining the REV of translucent porous media and inner contaminant based on two sand-box experiments. This paper is interesting, however, some details are missing. So I suggest "Major revision". My comments are as follows. (1) In the abstract, the new method of determining criterion should be pointed out clearly. (2) Light transmission techniques are very useful in two experiments. As shown in Eqs. (1)-(5), some parameters are important, but these parameters are not introduced in the following experiments and analysis. (3) In Lines 141-142, an assumption that the particles and pores are with lamellar structure is made. Further explanation and justification should be made for the reasonability of the assumption. (4) From Figure 7, the pattern of minimum REV sizes of porosity, sand density and torturosity is

quite different. Further explanation should be given based the new criterion. (5) The innovative point of this paper lies in the proposed criterion of determining REV. Two experiments have been carried out to validate the accuracy and reasonability of the criteria. However, the applicability of this method still requires to be further validated and clarified, because two cases are not enough and scale effects exists. (6) The mean size of REV is made based on its relations with porosity, density and tortuosity. Other variables, such as pressure or saturation, can be served as an additional indicator? Minor comments: (1) In Line 14, what are "previous REV estimation"? (2) In Line 15, a new criterion should be clarified. (3) In Line 23, cannot ? (4) In Line 51-52, Fig.1c is cited before Fig.1a and 1b. (5) In Line 119, Table1 should be "Table 1". (6) In Lines 217-218, the sub and sup i should be consistent. (7) In Line 552, volume? (8) In Table 1, how do you know permeability of the sand? (9) In Line 623, the subtitle of Fig.5a can be confusing, and it is suggested to replace porosity, density and tortuosity with other words.

---

## Referee Comment (RC2) · Anonymous Referee #2 · 2 Jul 2020

This paper proposes a new criterion for identifying representative elementary volume (REV) of translucent silica sands. Two sandbox experiments were conducted to test the applicability of the proposed criterion. The authors stated that the proposed criterion is effective and reliable. However, there are some important issues in the current manuscript that should be resolved or addressed.

Major comments:

1. The authors have published a series of paper on this topic. The research gap and the reason why a new criterion for REV is need should be clearly stated in the Introduction section.

[Figure]

2. Is the proposed criterion purely empirical or with some physical basis? If it is a criterion with physical basis, then the physical basis or the derivation process should also be added.

3. The blue curve of II-1 in the last figure of Fig. 4 is totally different from other curves. For other curves, the blue line becomes zero when the read line is zero. But for this figure, the blue line has a very big peak when the red line becomes zero. So the results of this figure are totally different from other figures. Such results seem does not support the authors' conclusion that "...is more convenient and reliable than other methods for REV estimation" in Lines 315-316.

4. The authors stated that "All observation cells show similar variation curves of ... that low value intervals are quite apparent, indicating that ... is vary effective to make the REV plateau obvious...", but it is not the case for the last figure in Figure 4b. As very different curves are obtained for Experiments I and II, it should be doubted that whether the new criterion is effective or not. Although the REV plateau may be identified based on the other figures in this study, but it is possibly that the REV plateau cannot easily be identified in other similar studies or in real porous materials.

5. The fit to cumulative frequency in Figure 5b is not very good. Both underestimation and overestimation exist.

6. Can the proposed criterion be applied to real world porous materials? Is the proposed criterion only applicable to the translucent silica sand used in this manuscript? The authors stated that fluid migration and transformation in porous media can be accurately simulated using the light transmission technique and the proposed criterion. Should the proposed criterion be used with the light transmission technique simultaneously? If yes, then the applicability of the proposed criterion is restricted to a very narrow range.

Minor comments: 1. Line 51: The authors used n to represent porosity, but then they used to represent porosity in Line 145. The authors again used n to represent porosity

in Line 148 Equation (5). 2. Line 127: What are the variation ranges of i and j in Equation (1)? They should be added to the equation. 3. Line 134: Add references to Equation (2) 4. Line 142: Add references to Equations (3) and (4) 5. Line 149: The quantity Ls seems not defined 6. Line 169: Is the "Hsies" should be "Hsieh"? 7. Lines 175-176: Reputation: "the derivative... will tend to zero" 8. Lines 176-177: References should be added to this sentence. 9. Line 182: Cannot find i in Figure 1b 10. Line 194: Here is , in Equation (11) is , which one is correct? 11. Lines 217-218: The authors should carefully check whether i should be in subscript or superscript. 12. Lines 218-220: Double check whether or should be used. 13. Line 238: Cannot find t=1.44 min in Figure 3b. 14. Lines 239-240: There should be error in this sentence or grammatical error 15. Line 243: There is no Fig. 2c 16. Line 253: Should be "Figs. 4a and b" 17. Line 269: There is no Fig. 4f, only Fig. 4a and 4b in this figure. 18. Line 338: Use a different symbol in Equation (16), because has already been used in Equation (15). 19. Line 358 and 359: Both are Experiment II? 20. Line 618: The subscripts and superscripts in the axis titles of Figure 4 can not be clearly seen 21. The equations listed in Table 2 are already included in the main text as Equations (10), (11), (14), and (15). Table 2 should be deleted. Also delete the citations and descriptions on Table 2. 22. I would suggest the authors modifying the numbers of figures and make sure the figure numbers appear in order in the text. For example, the authors first cited Fig. 1c in Line 52 and then Fig. 1a in Line 96 and Fig. 1b in Line 140. Generally, we should fist cite Fig. 1a, then Fig. 1b, and then Fig. 1c in order. 23. Table 3: Delete the equations and just list the parameter values.

---

## Referee Comment (RC3) · Anonymous Referee #3 · 24 Jul 2020

Review on "A new criterion for determining the representative elementary volume of translucent porous media and inner contaminant"

Wu et al. proposed a new criterion to determine the representative elementary volume (REV) of translucent porous media and inner contaminant, compared the new criterion with previous methods in two sandbox experiments, used the new criterion to calculate REVs of PCE plume (such as saturation, PCE-water interfacial area), and analyzed the influence of saturation on the REVs of saturation and PCE-water interfacial area. Although I do see some improvements of the new criterion in the Figure 4, the current paper is not suitable for the publication in HESS journal and needs major revision.

Detailed comments are as follows.

Major comments: (1) The title of the paper emphasizes on the new criterion, but only Figure 4 shows the comparison between the new criterion and other methods. Why do you design the new criterion as the current form? Why the new criterion has such improvements compared with other methods? These need to be introduced and discussed.

(2) Half part of the paper focuses on the "REVs of material properties" and "REVs of So and AOW for PCE plume", but there is no introduction on those topics in the "introduction" section. This makes it confusing on the contribution of this paper as compared with previous research.

(3) The experimental design is not introduced clearly. For example, why do you use two sandboxes with different materials? Why do the two sandboxes have different size? How to observe different variables with different cuboid window scale? Moreover, I think the method and result are mixed in the current paper. For example, L241-251 and L364-373 are methods instead of the results, so the author should move them to the section 2 to clarify the whole procedure you performed.

(4) The figure organization makes the paper not easy to follow. Figures are introduced from Fig. 1c to Fig. 1a, then to Figs. 2a-b, then back to Fig. 1b. I suggest the author to reorganize the figures just as the orders they appear in the paper.

(5) Figure 4. I see the difference of REV determined by "the relative gradient error" and "the new criterion method", which one we should trust? How to approve that the REV calculated by new criterion method is more reliable? Moreover, you can highlight the REV region in Figure 4 so that readers can directly see that.

(6) Figure 6. There is not any interpret or discussion on the Figure 6. If the figure is important, please provide detail description. If not, I suggest moving it to the supplementary.

[Figure]

(7) L383-384. In the downright corner of the Figure 7a, the red line increases first, then decrease. So I do not agree with that "while REV of PCE plume presents apparent decreasing ... for Experiment-II".

Minor comments: (1) L54, "As measured scale size ranging between Lmin and Lmax," Please give the Lmin and Lmax directly in the figure.

(2) Is there any reference for the conceptual representation of "REV curve" in L50?

(3) L142. "Fig. 1c" should be "Fig. 1d".

(4) L148. What does "n" mean in the Equation 5? And, the porosity does not occur in the Equation, how do you derive the porosity from it?

(5) L218-220. What is the difference between the and ? Are they the same?

(6) The author should proofread the paper carefully, as the current paper has numerous typos. For example, L243: "Figure 2c" cannot be found in the paper. L358, "All mean REV sizes of these variables for Experiment-II are larger than REVs of Experiments-II". L386-387, the sentence does not have verb.

---

## Author Comment (AC1) · 21 Sep 2020

Note that the following text in Arial Narrow font denotes Editor's and Reviewers' comments and in Times New Roman font denotes our response to the comments in the review. In our resubmission, the marked PDF file (Wu\_et\_al\_R1\_marked.pdf) has clearly indicated all changes to the original manuscript. Also, in our marked PDF file, marked in a green strikethrough font is the text that should be removed from the original manuscript and marked in a red font is the text that has been added to the revision. In addition, Line number(s) mentioned below is referred to as that line numbering in the marked revised manuscript.

Response to Anonymous Referee #2's CommentsiijŽ Anonymous Referee #2 Received and published: 2 July 2020 This paper proposes a new criterion for identifying representative elementary volume (REV) of translucent silica sands. Two sandbox experiments were conducted to test the applicability of the proposed criterion. The authors stated that the proposed criterion is effective and reliable. However, there are some important issues in the current manuscript that should be resolved or addressed. [Response] Comments accepted. We appreciate Referee #2's positive comments. We have fully addressed the concerns raised by Referee #2's in the revised manuscript to improve the manuscript and given a point-to-point response to the reviewer's comments as below. Major comments: 1. The authors have published a series of paper on this topic. The research gap and the reason why a new criterion for REV is need should be clearly stated in the Introduction section. [Response] Comments accepted. We have added associated description into the Introduction section (Lines 90-93). 2. Is the proposed criterion purely empirical or with some physical basis? If it is a criterion with physical basis, then the physical basis or the derivation process should also be added. [Response] Comments accepted. The new criterion conforms to the Eq. (12). Moreover, the new criterion is proposed based on the dimensionless range () (Brown and Hsieh, 2000). However, is hard to be achieved. According to the [Eq. (12)], we propose a new criterion and test the effect for translucent porous media. 3. The blue curve of II-1 in the last figure of Fig. 4 is totally different from other curves. For other curves, the blue line becomes zero when the read line is zero. But for this figure, the blue line has a very big peak when the red line becomes zero. So the results of this figure are totally different from other figures. Such results seem does not support the authors' conclusion that "...is more convenient and reliable than other methods for REV estimation" in Lines 315-316. [Response] Comments accepted. Referee #2 may refer to the So- $\chi i$  of II-1. The blue line becomes zero when L

the So-REVs of II-1 and II-2 have different values. By the help of the new criterion, REV estimation is more convenient. To make zero part of blue line more apparent, the blue line is thickened in the last figure of Fig. 4. We have used open circles to indicate the REV plateau region in Fig. 4. Readers can see REV plateau estimated by the new criterion. (Lines 697-700) 4. The authors stated that "All observation cells show similar variation curves of ... that low value intervals are quite apparent, indicating that ... is vary effective to make the REV plateau obvious...", but it is not the case for the last figure in Figure 4b. As very different curves are obtained for Experiments I and II, it should be doubted that whether the new criterion is effective or not. Although the REV plateau may be identified based on the other figures in this study, but it is possibly that the REV plateau cannot easily be identified in other similar studies or in real porous materials. [Response] Comments accepted. Curves all have low value intervals in Fig. 4 for So- $\chi$ i, so we treat these curves as similar variation curves. In the last figure in Fig. 4b, the low value interval of blue line is not apparent, so the blue line is thickened to make the low value interval apparent. (Lines 697-700) 5. The fit to cumulative frequency in Figure 5b is not very good. Both underestimation and overestimation exist. [Response] Comments accepted. We have made effect to improve the fit to cumulative frequency in Fig. 5b (Lines 703-705). 6. Can the proposed criterion be applied to real world porous materials? Is the proposed criterion only applicable to the translucent silica sand used in this manuscript? The authors stated that fluid migration and transformation in porous media can be accurately simulated using the light transmission technique and the proposed criterion. Should the proposed criterion be used with the light transmission technique simultaneously? If yes, then the applicability of the proposed criterion is restricted to a very narrow range. [Response] Comments accepted. We appreciate the reviewer's insightful comment. In this study, we only focus on characterizing the REV of translucent silica sand and inner PCE plume at lab scale based on light transmission technique. Due to multiple limitations of x-ray and gramma ray causing high cost, inefficiency, complex high energy sources and hazard environment in materials measurements, light transmission technique is used to achieve rapid.
hand and economical measurements of materials with high resolution and good effectiveness. However, minimum REV size (Lmin) and maximum REV size (Lmax) can't be identified simultaneously for translucent silica sand based on previous criteria and light transmission technique. So this new criterion is proposed to improve the effect of REV estimation for translucent silica sand. In this study, the proposed criterion is used with the light transmission technique. However, we believe its potential applicability can't be treated as a narrow range by this study. We think this issue is beyond this study and the applicability of the new criterion will be explored in our further work. Minor comments: 1. Line 51: The authors used n to represent porosity, but then they used to represent porosity in Line 145. The authors again used n to represent porosity in Line 148 Equation (5). [Response] Comments accepted. We have replaced 'n' with ' $\theta$ ' (Lines 53 and 158). 2. Line 127: What are the variation ranges of i and j in Equation (1)? They should be added to the equation. [Response] Comments accepted. We have used a and b in Eq.(1) to represent phase number and interface number (Lines 136-138). 3. Line 134: Add references to Equation (2) [Response] Comments accepted. We have added references to Equation (2) (Lines 143-144). 4. Line 142: Add references to Equations (3) and (4) [Response] Comments accepted. We have added reference to Equations (3) and (4) (Line 152). 5. Line 149: The quantity Ls seems not defined [Response] Comments accepted. We have checked carefully and made correction (Line 159). 6. Line 169: Is the "Hsies" should be "Hsieh"? [Response] Comments accepted and correction made (Line 205). 7. Lines 175-176: Reputation: "the derivative... will tend to zero" [Response] Comments accepted. We have deleted "the derivative... will tend to zero" (Lines 211-212). 8. Lines 176-177: References should be added to this sentence. [Response] Comments accepted. We have added references (Line 214-215). 9. Line 182: Cannot find i in Figure 1b [Response] Comments accepted. The cuboid window is presented in Fig. 1b, i refers to the window increment number. We have modified the numbers of figures in Fig. 1 (Lines 653-659, 683-685). 10. Line 194: Here is , in Equation (11) is , which one is correct? [Response] Comments accepted. The cuboid window is presented in Fig. 1b, i refers to the window
increment number. m(i) is the total number of sub-grids in measured cuboid window. 11. Lines 217-218: The authors should carefully check whether i should be in subscript or superscript. [Response] Comments accepted. We have corrected the sub i to sup i (Line 256). 12. Lines 218-220: Double check whether or should be used. [Response] Comments accepted. We have checked carefully and corrected the subscripts and superscripts (Line 256). 13. Line 238: Cannot find t=1.44 min in Figure 3b. [Response] Comments accepted. We have corrected this mistake (Line 278). 14. Lines 239-240: There should be error in this sentence or grammatical error [Response] Comments accepted. We have revised this sentence (Lines 278-280). 15. Line 243: There is no Fig. 2c [Response] Comments accepted. We have deleted "Fig. 2c" (Line 180). 16. Line 253: Should be "Figs. 4a and b" [Response] Comments accepted. We have made correction (Line 293). 17. Line 269: There is no Fig. 4f, only Fig. 4a and 4b in this figure. [Response] Comments accepted. We have replaced "Figs. 4a-f" to "Figs. 4a and b" (Line 310). 18. Line 338: Use a different symbol in Equation (16). because has already been used in Equation (15). [Response] Comments accepted. We have used a different symbol (Lines 377-378). 19. Line 358 and 359: Both are Experiment II? [Response] Comments accepted. All mean REV sizes of these variables for Experiment-II are larger than REVs of Experiments-I. We have made corresponding correction (Line 402). 20. Line 618: The subscripts and superscripts in the axis titles of Figure 4 can not be clearly seen [Response] Comments accepted. We have revised Figure 4 to make subscripts and superscripts clearer (Lines 697-700). 21. The equations listed in Table 2 are already included in the main text as Equations (10). (11). (14), and (15). Table 2 should be deleted. Also delete the citations and descriptions on Table 2. [Response] Comments accepted. We have deleted Table and associated citations and descriptions (Lines 627-628). 22. I would suggest the authors modifying the numbers of figures and make sure the figure numbers appear in order in the text. For example, the authors first cited Fig. 1c in Line 52 and then Fig. 1a in Line 96 and Fig. 1b in Line 140. Generally, we should fist cite Fig. 1a, then Fig. 1b, and then Fig. 1c in order. [Response] Comments accepted. We have modified the numbers of figures
in Fig. 1 (Lines 653-659, 683-685). 23. Table 3: Delete the equations and just list the parameter values. [Response] Comments accepted. We have deleted the equations and list parameter values in Table 2 (Lines 630-634).

Response to Anonymous Referee #3's CommentsïijŽ Review on "A new criterion for determining the representative elementary volume of translucent porous media and inner contaminant" Wu et al. proposed a new criterion to determine the representative elementary volume (REV) of translucent porous media and inner contaminant, compared the new criterion with previous methods in two sandbox experiments, used the new criterion to calculate REVs of PCE plume (such as saturation, PCE-water interfacial area), and analyzed the influence of saturation on the REVs of saturation and PCE-water interfacial area. Although I do see some improvements of the new criterion in the Figure 4, the current paper is not suitable for the publication in HESS journal and needs major revision. [Response] We appreciate Referee #3's positive comments. Also, we have fully addressed the issues raised by the reviewer and made major revision in the revised manuscript, and given a point-to-point response to the reviewer's comments as follows.

Detailed comments are as follows. Major comments: (1) The title of the paper emphasizes on the new criterion, but only Figure 4 shows the comparison between the new criterion and other methods. Why do you design the new criterion as the current form? Why the new criterion has such improvements compared with other methods? These need to be introduced and discussed. [Response] Comments accepted. We have added more expression into the Introduction section. The new criterion conforms to the Eq. (12). Moreover, the new criterion is proposed based on the dimensionless range () (Brown and Hsieh, 2000). However, is hard to be achieved. According to the [Eq. (12)], we propose a new criterion and test the effect for translucent porous media. The results suggest the new criterion appears to be the most appropriate criterion for REV plateau identification (Lines 90-93, 253). (2) Half part of the paper focuses on the "REVs of material properties" and "REVs of So and AOW for PCE plume",
but there is no introduction on those topics in the "introduction" section. This makes it confusing on the contribution of this paper as compared with previous research. [Response] Comments accepted. We have added REVs of material properties and PCE saturation, PCE-water interfacial area in the introduction section (Lines 71-72, 84-88). (3) The experimental design is not introduced clearly. For example, why do you use two sandboxes with different materials? Why do the two sandboxes have different size? How to observe different variables with different cuboid window scale? Moreover, I think the method and result are mixed in the current paper. For example, L241-251 and L364-373 are methods instead of the results, so the author should move them to the section 2 to clarify the whole procedure you performed. [Response] Comments accepted. We have added a heterogeneous case (Experiment-III) to validate the applicability of new criteria for REV estimation. The methods are moved to the section 2 (Lines 176-201). (4) The figure organization makes the paper not easy to follow. Figures are introduced from Fig. 1c to Fig. 1a, then to Figs. 2a-b, then back to Fig. 1b. I suggest the author to reorganize the figures just as the orders they appear in the paper. [Response] Comments accepted. We have modified the numbers of figures in Fig. 1 (Lines 653-659, 683-685). (5) Figure 4. I see the difference of REV determined by "the relative gradient error" and "the new criterion method", which one we should trust? How to approve that the REV calculated by new criterion method is more reliable? Moreover, you can highlight the REV region in Figure 4 so that readers can directly see that. [Response] Comments accepted. The relative gradient error is proposed by previous study and has also used for our research about translucent porous media and contaminants migration. However, random fluctuations exist in curves visually, which make the REV plateau uneasy to be identified. Significantly, the curve of new criterion appears low value interval which makes the beginning and ending of REV plateau easier to be identified. We have used open circles to indicate the REV plateau region in Fig. 4. Readers can see REV plateau estimated by the new criterion. (6) Figure 6. There is not any interpret or discussion on the Figure 6. If the figure is important, please provide detail description. If not, I suggest moving

**HESSD**
it to the supplementary. [Response] Comments accepted. Fig.7 is obtained on the REV distribution presented in Fig. 6. We have added more discussion about Fig.6 in revised manuscript (Lines 407-419). (7) L383-384. In the downright corner of the Figure 7a, the red line increases first, then decrease. So I do not agree with that "while REV of PCE plume presents apparent decreasing ... for Experiment-II". [Response] Comments accepted. We have revised this sentence in revised manuscript (Lines 447-448). Minor comments: (1) L54, "As measured scale size ranging between Lmin and Lmax," Please give the Lmin and Lmax directly in the figure. [Response] Comments accepted. We have added "Lmin" and "Lmax" in Fig. 1a (Lines 683-685). (2) Is there any reference for the conceptual representation of "REV curve" in L50? [Response] Comments accepted. We have added reference for "REV curve" (Lines 52-53). (3) L142. "Fig. 1c" should be "Fig. 1d". [Response] Comments accepted. We have made corresponding correction (Line 152). (4) L148. What does "n" mean in the Equation 5? And, the porosity does not occur in the Equation, how do you derive the porosity from it? [Response] Comments accepted. We have replaced 'n' with ' $\theta$ ' (Lines 53 and 158). (5) L218-220. What is the difference between the and? Are they the same? [Response] Comments accepted. We have corrected the sub and sup i (Line 255). (6) The author should proofread the paper carefully, as the current paper has numerous typos. For example, L243: "Figure 2c" cannot be found in the paper. L358, "All mean REV sizes of these variables for Experiment-II are larger than REVs of Experiments-II". L386-387, the sentence does not have verb. [Response] Comments accepted. We have checked carefully and corrected these mistakes above (Lines 180, 402 and 431-433).

Please also note the supplement to this comment: https://hess.copernicus.org/preprints/hess-2020-91/hess-2020-91-AC1supplement.zip

91, 2020.
Fig. 1

**Fig. 1.** (a) Variable changes as measured scale (L) increment in conceptual curve (Costanza-Robinson et al., 2011); (b) Scale effect and the cuboid image window geometry; (c) System Device for acquisition of p

35
**HESSD**
Fig. 2. (a) The system sandbox equipment of Experiment-I; (b) The system sandbox equipment of Experiment-II; (c) The system sandbox equipment of Experiment-III

36

37

Fig. 3. (a) The emergent light intensity, porosity, permeability and tortuosity of 2D translucent

silica sand for Experiments-I-III; (b) The PCE saturation of Experiments-I-III during 0~1523 min

2.32 2.28 2.24 2.20 2.16

---

## Author Comment (AC3) · 21 Sep 2020

Response to Referee #1: Please see the attached PDF file "Response_to_HESS_Referee1_R1.pdf" in which we have given a point-by-point response to Reviewers' comments. Note that the following text in Arial Narrow font denotes Editor's and Reviewers' comments and in Times New Roman font denotes our response to the comments in the review. In our resubmission, the marked PDF file (Wu_et_al_R1_marked.pdf) has clearly indicated all changes to the original manuscript. Also, in our marked PDF file, marked in a green strikethrough font is the text that should be removed from the original manuscript and marked in a red font is

the text that has been added to the revision. In addition, Line number(s) mentioned below is referred to as that line numbering in the marked revised manuscript.

Please also note the supplement to this comment:
https://hess.copernicus.org/preprints/hess-2020-91/hess-2020-91-AC3-supplement.zip

———————————————————

Fig. 1

[Figure]

35

**Fig. 1.** (a) Variable changes as measured scale (L) increment in conceptual curve (Costanza-Robinson et al., 2011); (b) Scale effect and the cuboid image window geometry; (c) System Device for acquisition of p

[Figure]

**Fig. 2.** (a) The system sandbox equipment of Experiment-I; (b) The system sandbox equipment of Experiment-II; (c) The system sandbox equipment of Experiment-III

Fig. 3

[Figure]

**Fig. 3.** (a) The emergent light intensity, porosity, permeability and tortuosity of 2D translucent silica sand for Experiments-I-III; (b) The PCE saturation of Experiments-I-III during 0~1523 min and observati

[Figure]

38

**Fig. 4.** (a) The change of porosity ($\theta$), associated coefficient of variation (C_Vˆi), entropy dimension (DIˆi), the relative gradient error ($\varepsilon$_gˆi), and new criterion-$\chi$i for observation cells as cuboid window

[Figure]

**Fig. 5.** (a) The distributions of minimum REV sizes of porosity, sand density and tortuosity for Experiments-I-III; (b) The frequency of minimum REV sizes of Experiments and fitted models

Fig. 6

[Figure]

**Fig. 6.** (a) The distributions of So-REV sizes during 0∼1523 min for Experiments-I-III; (b) The distributions of AOW-REV sizes during 0∼1523 min for Experiments-I-III

Fig. 7

[Figure]

**Fig. 7.** (a) The mass center coordinate of PCE plume, GTP, plume area and the mean, standard deviation of So-REV and AOW-REV during 0∼1523 min; (b) The change of average REV size as the distance dl, dm increas

---

## Author Response (AR1)

Response to Editor's and Reviewers' Comments on Manuscript hess-2020-91 "A new criterion for determining the representative elementary volume of translucent porous media and inner contaminant" by Ming Wu, Jianfeng Wu, Jichun Wu, and Bill X. Hu

Note that the following text in Arial Narrow font denotes Editor's and Reviewers' comments and in Times New Roman font denotes our response to the comments in the review. In our Supplement (pdf/zip), the PDF file has clearly indicated all changes to the original manuscript. In our Manuscript (pdf), all changes have been included in the clean version of the revised PDF file. Also, in our marked PDF file, marked in  is the text that should be removed from the original manuscript and marked in a red font is the text that has been added to the revision. In addition, Line number(s) mentioned below is referred to as that line numbering in the marked revised manuscript.

**Response to Anonymous Referee #1's Comments**

Anonymous Referee #1

This paper presents a study of determining the REV of translucent porous media and inner contaminant based on two sand-box experiments. This paper is interesting, however, some details are missing. So I suggest "Major revision".

**[Response]** Comments accepted. We appreciate Referee #1's conscientious and positive recommendation. We have fully addressed Referee #1's concerns in the revised manuscript.

My comments are as follows.

(1) In the abstract, the new method of determining criterion should be pointed out clearly.

**[Response]** Comments accepted. We have revised this sentence to point out the new criteria (Lines 14-16).

(2) Light transmission techniques are very useful in two experiments. As shown in Eqs. (1)-(5), some parameters are important, but these parameters are not introduced in the following experiments and analysis.

**[Response]** Comments accepted. We have added a heterogeneous case (Experiment-III) to validate the applicability of new criteria for REV estimation.

To derive two fitting constants, some procedures should be taken. For heterogeneous case (Experiment-III), six grades of commercial translucent Accusand silica sand were used to pack the sandbox. Background material was packed by the F20/30 mesh sand and F70/F100, F70/F80, F40/F60, F50/F70, F35/F50 mesh sands with low permeability were used to pack five lenses (Fig. 2c and Table A1).

Table A1 Properties of six kinds of translucent silica sand

| Property of sands | F20/30 background | F30/40 Lens E | F50/70 Lens D | F40/50 Lens B | F70/80 Lens C | F70/100 Lens A |
|---|---|---|---|---|---|---|
| Median grain diameter (cm) [a] | 0.072 | 0.036 | 0.026 | 0.034 | 0.022 | 0.016 |
| Porosity [b] | 0.331 | 0.304 | 0.249 | 0.277 | 0.221 | 0.201 |
| Permeability (m$^2$) [a] | $1.35 \times 10^{-10}$ | $8.85 \times 10^{-11}$ | $3.66 \times 10^{-11}$ | $6.38 \times 10^{-11}$ | $8.19 \times 10^{-12}$ | $4.69 \times 10^{-12}$ |
| Entry pressure (kPa) [a] | 0.049 | 0.203 | 1.058 | 0.490 | 2.048 | 3.895 |

[a] refer to O'Carroll et al. (2004)

[b] from experiment measurement and refer to Bradford et al. (1999)

Temporal emergent light intensity distribution before PCE injection into sandbox was collected as in Fig. 3a which every pixel is 0.482mm ×0.523mm. Obviously, area of every pixel approaches zero and obey the requirement of Light transmission technique. The average emergent light intensity of lenses A, B, C, D, E and background material F20/30 mesh sand are derived from Fig. 3a and their porosity is listed in Table A1. The relationship between light intensity and porosity developed by Light transmission technique as Eqs. (1)-(5) is validated in Fig. A1. There is a fairly agreement between light intensity and porosity with $R^2$ value equals to 0.9818 that any significant bias doesn't appear in the validation results (Fig. A1). The parameters γ and β in Eq. (5) are achieved from validation results and the porosity distribution of 2D porous media achieved by Light transmission technique is shown in Fig. A1. The whole mass of sand packed in the sandbox is calculated:

$$M_c = \sum_{i_2}^{m_2} \sum_{i_1}^{m_1} (1 - \theta_{i_1,i_2}) L_1 L_2 L_3 \rho_s \tag{A1}$$

where $M_c$ is the total mass of sand calculated from Light transmission technique; $i_1$ is the layer number of computing grid; $i_2$ is the row number of computing grid; $m_1$ is total number of layers; $m_2$ is the total number of rows; $\theta_{i,j}$ is the porosity of computing grid; $L_1$ is the length of computing grid; $L_2$ is the width of computing grid; $L_3$ is the thickness of computing grid; $\rho_s$ is the density of sand particles. In comparison with the actual mass of sand in experiment, the relative error of 4.85% is achieved by Light transmission technique for calculation of sand mass. These results indicate a good agreement between the quantifications by Light transmission technique and experiment observations.

[Figure]

Fig. A1 The emergent light intensity versus porosity for different mesh sands used in experiment.

For other experiments, the emergent light intensity is corrected using the method expressed by Bob et al. (2008). All images collected were automatically corrected for the dark signal (baseline) associated with the CCD detector by subtracting an image taken at the same exposure time with the camera shutter closed. To correct for the inevitable temporal variation in light intensity, a small region, referred to as a ''correction box'', in one of the collected images (referred to as the reference image) was identified. The reference image was chosen to be the image collected when the model was fully saturated with water and was packed by background material (F20/30 mesh sand) for Experiment-III. In choosing the correction box, it was very important to make sure that, for all other images of Experiments-I and II, this region always remained under the same conditions of full water saturation and same mesh sand. Thus, any change in light intensity within the correction box was due to changes in source light intensity. The average light intensity of this correction box was calculated for all images including the reference image. To correct a particular image of other experiments, light intensities from image were simply multiplied by the ratio of the average light intensity of the correction box for the reference image and the average light intensity of the correction box for the image to be corrected.

Afterward, porosity is quantified by Eq. (5). Then density and tortuosity can be achieved using Eqs. (6) and (7).

(3) In Lines 141-142, an assumption that the particles and pores are with lamellar structure is made. Further explanation and justification should be made for the reasonability of the assumption.

[Response] Comments accepted. To quantify the porosity of translucent silica sand by light transmission method, we suppose 2D translucent silica sand is consist of various infinitesimal elements whose area approaches zero. An infinitesimal element is selected from 2D translucent silica sand which area approaches zero (Fig. 1d). The size of cross-sectional area of infinitesimal element is less than the particle size of silica sand. Therefore, the infinitesimal element can be treated as lamellar structure shown in Fig. 1d. Obviously, area of every pixel approaches zero and obey the requirement of Light transmission technique. In comparison with the actual mass of sand in experiment, the relative error of 4.85% is achieved by Light transmission technique for calculation of sand mass.

(4) From Figure 7, the pattern of minimum REV sizes of porosity, sand density and torturosity is quite different. Further explanation should be given based the new criterion.

**[Response]** Comments accepted. The materials packed in two sandboxes are different. F40/50 and F20/30 mesh translucent silica sands are used for Experiments-I and II. So the minimum REV sizes of two experiments are different. What's more, the REV sizes of different parameters have different patterns. The REVs of porosity, moisture saturation ($S_W$) and interfacial area ($A_I$) also obtained different values according to Costanza-Robinson et al. (2011). The relationship observed for $S_W$-REV and $S_W$ are different from the relationship between $A_I$-REV and $S_W$. Therefore, the REV of different parameter is possible to be different.

(5) The innovative point of this paper lies in the proposed criterion of determining REV. Two experiments have been carried out to validate the accuracy and reasonability of the criteria. However, the applicability of this method still requires to be further validated and clarified, because two cases are not enough and scale effects exists.

**[Response]** Comments accepted. We have added a heterogeneous case (Experiment-III) into the revised manuscript (Fig. 2c).

(6) The mean size of REV is made based on its relations with porosity, density and tortuosity. Other variables, such as pressure or saturation, can be served as an additional indicator?

**[Response]** Comments accepted. The REV of porous media is made based on its relations with porosity, density and tortuosity. However, PCE saturation is an indicator for PCE plume in porous media, the REV of PCE saturation can be used to obtain the REV of PCE plume.

Minor comments:
(1) In Line 14, what are "previous REV estimation"?

**[Response]** Comments accepted. We have corrected "previous REV estimation criteria" to "existing REV estimation criteria" (Lines 14-15).

(2) In Line 15, a new criterion should be clarified.

**[Response]** Comments accepted. We have made correction to clarify the new criterion (Lines 14-17).

(3) In Line 23, cannot ?

**[Response]** Comments accepted. We have replaced 'can not' with 'are not effective' (Line 25).

(4) In Line 51-52, Fig.1c is cited before Fig.1a and 1b.

**[Response]** Comments accepted. We have modified the numbers of figures in Fig. 1 (Lines 653-659, 683-685).

(5) In Line 119, Table1 should be "Table 1".

**[Response]** Comments accepted. We have corrected "Table1" to "Table 1" (Line 128).

(6) In Lines 217-218, the sub and sup i should be consistent.

**[Response]** Comments accepted. We have corrected the sub and sup i (Line 256).

(7) In Line 552, volume?

**[Response]** Comments accepted and correction made accordingly (Line 618).

(8) In Table 1, how do you know permeability of the sand?

**[Response]** Comments accepted. The average permeability of silica sand is obtained by experiment and research references. Moreover, accurate permeability of 2D translucent silica sand can be calculated by the help of light transmission technique and fractal method. Afterward, average permeability of silica sand also can be obtained.

(9) In Line 623, the subtitle of Fig.5a can be confusing, and it is suggested to replace porosity, density and tortuosity with other words.

**[Response]** Comments accepted. We have made correction according to suggestion (Lines 703-705).

**Response to Anonymous Referee #2's Comments:**

Anonymous Referee #2

This paper proposes a new criterion for identifying representative elementary volume (REV) of translucent silica sands. Two sandbox experiments were conducted to test the applicability of the proposed criterion. The authors stated that the proposed criterion is effective and reliable. However, there are some important issues in the current manuscript that should be resolved or addressed.

**[Response]** Comments accepted. We appreciate Referee #2's positive comments. We have fully addressed the concerns raised by Referee #2's in the revised manuscript to improve the manuscript and given a point-to-point response to the reviewer's comments as below.

Major comments:

1. The authors have published a series of paper on this topic. The research gap and the reason why a new criterion for REV is need should be clearly stated in the Introduction section.

**[Response]** Comments accepted. We have added associated description into the Introduction section (Lines 90-93).

2. Is the proposed criterion purely empirical or with some physical basis? If it is a criterion with physical basis, then the physical basis or the derivation process should also be added.

**[Response]** Comments accepted. The new criterion conforms to the Eq. (12). Moreover, the new criterion is proposed based on the dimensionless range ($\delta^i$) (Brown and Hsieh, 2000). However, $\delta^i \ll 1$ is hard to be achieved. According to the $\frac{\partial Y(L_i)}{\partial L}\big|_{L_i = L_o} = 0$ [Eq. (12)], we propose a new criterion $\chi^i = \frac{|\delta^{i+1} - \delta^{i-1}|}{\delta^i \Delta L}$ and test the effect for translucent porous media.

3. The blue curve of II-1 in the last figure of Fig. 4 is totally different from other curves.
For other curves, the blue line becomes zero when the read line is zero. But for this figure, the blue line has a very big peak when the red line becomes zero. So the results of this figure are totally different from other figures. Such results seem does not support the authors' conclusion that "...is more convenient and reliable than other methods for REV estimation" in Lines 315-316.

**[Response]** Comments accepted. Referee #2 may refer to the $S_o$-$\chi^i$ of II-1. The blue line becomes zero when L<5.0mm, suggest the REV size of $S_o$ for II-1 has small value compared to II-2. In the last figure of Fig. 4, the blue line first becomes zero when red line has large value. As scale increases, the red line becomes zero while the blue line has large value again. The phenomenon suggests the $S_o$-REVs of II-1 and II-2 have different values. By the help of the new criterion, REV estimation is more convenient. To make zero part of blue line more apparent, the blue line is thickened in the last figure of Fig. 4. We have used open circles to indicate the REV plateau region in Fig. 4. Readers can see REV plateau estimated by the new criterion. (Lines 697-700)

4. The authors stated that "All observation cells show similar variation curves of ... that low value intervals are quite apparent, indicating that ... is vary effective to make the REV plateau obvious...", but it is not the case for the last figure in Figure 4b. As very different curves are obtained for Experiments I and II, it should be doubted that whether the new criterion is effective or not. Although the REV plateau may be identified based on the other figures in this study, but it is possibly that the REV plateau cannot easily be identified in other similar studies or in real porous materials.

**[Response]** Comments accepted. Curves all have low value intervals in Fig. 4 for $S_o$-$\chi^i$, so we treat these curves as similar variation curves. In the last figure in Fig. 4b, the low value interval of blue line is not apparent, so the blue line is thickened to make the low value interval apparent. (Lines 697-700)

5. The fit to cumulative frequency in Figure 5b is not very good. Both underestimation and overestimation exist.

**[Response]** Comments accepted. We have made effect to improve the fit to cumulative frequency in Fig. 5b (Lines 703-705).

6. Can the proposed criterion be applied to real world porous materials? Is the proposed criterion only applicable to the translucent silica sand used in this manuscript?
The authors stated that fluid migration and transformation in porous media can be accurately simulated using the light transmission technique and the proposed criterion.
Should the proposed criterion be used with the light transmission technique simultaneously?
If yes, then the applicability of the proposed criterion is restricted to a very narrow range.

**[Response]** Comments accepted. We appreciate the reviewer's insightful comment. In this study, we only focus on characterizing the REV of translucent silica sand and inner PCE plume at lab scale based on light transmission technique. Due to multiple limitations of x-ray and gramma ray causing high cost, inefficiency, complex high energy sources and hazard environment in materials measurements, light transmission technique is used to achieve rapid, hand and economical measurements of materials with high resolution and good effectiveness. However, minimum REV size ($L_{min}$) and maximum REV size ($L_{max}$) can't be identified simultaneously for translucent silica sand based on previous criteria and light transmission technique. So this new criterion is proposed to improve the effect of REV estimation for translucent silica sand. In this study, the proposed criterion is used with the light transmission technique. However, we believe its potential applicability can't be treated as a narrow range by this study. We think this issue is beyond this study and the applicability of the new criterion will be explored in our further work.

Minor comments:

1. Line 51: The authors used n to represent porosity, but then they used to represent porosity in Line 145. The authors again used n to represent porosity in Line 148 Equation (5).

**[Response]** Comments accepted. We have replaced 'n' with '$\theta$' (Lines 53 and 158).

2. Line 127: What are the variation ranges of i and j in Equation (1)? They should be added to the equation.

**[Response]** Comments accepted. We have used $a$ and $b$ in Eq.(1) to represent phase number and interface number (Lines 136-138).

3. Line 134: Add references to Equation (2)

**[Response]** Comments accepted. We have added references to Equation (2) (Lines 143-144).

4. Line 142: Add references to Equations (3) and (4)

**[Response]** Comments accepted. We have added reference to Equations (3) and (4) (Line 152).

5. Line 149: The quantity Ls seems not defined

**[Response]** Comments accepted. We have checked carefully and made correction (Line 159).

6. Line 169: Is the "Hsies" should be "Hsieh"?

**[Response]** Comments accepted and correction made (Line 205).

7. Lines 175-176: Reputation: "the derivative... will tend to zero"

**[Response]** Comments accepted. We have deleted "the derivative... will tend to zero" (Lines 211-212).

8. Lines 176-177: References should be added to this sentence.

**[Response]** Comments accepted. We have added references (Line 214-215).

9. Line 182: Cannot find i in Figure 1b

**[Response]** Comments accepted. The cuboid window is presented in Fig. 1b, i refers to the window increment number. We have modified the numbers of figures in Fig. 1 (Lines 653-659, 683-685).

10. Line 194: Here is , in Equation (11) is , which one is correct?

**[Response]** Comments accepted. The cuboid window is presented in Fig. 1b, i refers to the window increment number. m(i) is the total number of sub-grids in measured cuboid window.

11. Lines 217-218: The authors should carefully check whether i should be in subscript or superscript.

**[Response]** Comments accepted. We have corrected the sub i to sup i (Line 256).

12. Lines 218-220: Double check whether or should be used.

**[Response]** Comments accepted. We have checked carefully and corrected the subscripts and superscripts (Line 256).

13. Line 238: Cannot find t=1.44 min in Figure 3b.

**[Response]** Comments accepted. We have corrected this mistake (Line 278).

14. Lines 239-240: There should be error in this sentence or grammatical error

**[Response]** Comments accepted. We have revised this sentence (Lines 278-280).

15. Line 243: There is no Fig. 2c

**[Response]** Comments accepted. We have deleted "Fig. 2c" (Line 180).

16. Line 253: Should be "Figs. 4a and b"

**[Response]** Comments accepted. We have made correction (Line 293).

17. Line 269: There is no Fig. 4f, only Fig. 4a and 4b in this figure.

**[Response]** Comments accepted. We have replaced "Figs. 4a-f" to "Figs. 4a and b" (Line 310).

18. Line 338: Use a different symbol in Equation (16), because has already been used in Equation (15).

**[Response]** Comments accepted. We have used a different symbol (Lines 377-378).

19. Line 358 and 359: Both are Experiment II?

**[Response]** Comments accepted. All mean REV sizes of these variables for Experiment-II are larger than REVs of Experiments-I. We have made corresponding correction (Line 402).

20. Line 618: The subscripts and superscripts in the axis titles of Figure 4 can not be clearly seen

**[Response]** Comments accepted. We have revised Figure 4 to make subscripts and superscripts clearer (Lines 697-700).

21. The equations listed in Table 2 are already included in the main text as Equations (10), (11), (14), and (15). Table 2 should be deleted. Also delete the citations and descriptions on Table 2.

**[Response]** Comments accepted. We have deleted Table and associated citations and descriptions (Lines 627-628).

22. I would suggest the authors modifying the numbers of figures and make sure the figure numbers appear in order in the text. For example, the authors first cited Fig. 1c in Line 52 and then Fig. 1a in Line 96 and Fig. 1b in Line 140. Generally, we should fist cite Fig. 1a, then Fig. 1b, and then Fig. 1c in order.

**[Response]** Comments accepted. We have modified the numbers of figures in Fig. 1 (Lines 653-659, 683-685).

23. Table 3: Delete the equations and just list the parameter values.

**[Response]** Comments accepted. We have deleted the equations and list parameter values in Table 2 (Lines 630-634).

**Response to Anonymous Referee #3's Comments:**

Review on "A new criterion for determining the representative elementary volume of translucent porous media and inner contaminant"

Wu et al. proposed a new criterion to determine the representative elementary volume (REV) of translucent porous media and inner contaminant, compared the new criterion with previous methods in two sandbox experiments, used the new criterion to calculate REVs of PCE plume (such as saturation, PCE-water interfacial area), and analyzed the influence of saturation on the REVs of saturation and PCE-water interfacial area. Although I do see some improvements of the new criterion in the Figure 4, the current paper is not suitable for the publication in HESS journal and needs major revision.

**[Response]** We appreciate Referee #3's positive comments. Also, we have fully addressed the issues raised by the reviewer and made major revision in the revised manuscript, and given a point-to-point response to the reviewer's comments as follows.

Detailed comments are as follows.

Major comments:

(1) The title of the paper emphasizes on the new criterion, but only Figure 4 shows the comparison between the new criterion and other methods. Why do you design the new criterion as the current form? Why the new criterion has such improvements compared with other methods? These need to be introduced and discussed.

**[Response]** Comments accepted. We have added more expression into the Introduction section. The new criterion conforms to the Eq. (12). Moreover, the new criterion is proposed based on the dimensionless range ($\delta^i$) (Brown and Hsieh, 2000). However, $\delta^i \ll 1$ is hard to be achieved. According to the $\frac{\partial Y(L_i)}{\partial L}|_{L_i=L_o}=0$ [Eq. (12)], we propose a new criterion $\chi^i = \frac{|\delta^{i+1} - \delta^{i-1}|}{\delta^i \Delta L}$ and test the effect for translucent porous media. The results suggest the new criterion appears to be the most appropriate criterion for REV plateau identification (Lines 90-93, 253).

(2) Half part of the paper focuses on the "REVs of material properties" and "REVs of So and AOW for PCE plume", but there is no introduction on those topics in the "introduction" section. This makes it confusing on the contribution of this paper as compared with previous research.

**[Response]** Comments accepted. We have added REVs of material properties and PCE saturation, PCE-water interfacial area in the introduction section (Lines 71-72, 84-88).

(3) The experimental design is not introduced clearly. For example, why do you use two sandboxes with

different materials? Why do the two sandboxes have different size? How to observe different variables with different cuboid window scale? Moreover, I think the method and result are mixed in the current paper. For example, L241-251 and L364-373 are methods instead of the results, so the author should move them to the section 2 to clarify the whole procedure you performed.

**[Response]** Comments accepted. We have added a heterogeneous case (Experiment-III) to validate the applicability of new criteria for REV estimation. The methods are moved to the section 2 (Lines 176-201).

(4) The figure organization makes the paper not easy to follow. Figures are introduced from Fig. 1c to Fig. 1a, then to Figs. 2a-b, then back to Fig. 1b. I suggest the author to reorganize the figures just as the orders they appear in the paper.

**[Response]** Comments accepted. We have modified the numbers of figures in Fig. 1 (Lines 653-659, 683-685).

(5) Figure 4. I see the difference of REV determined by "the relative gradient error" and "the new criterion method", which one we should trust? How to approve that the REV calculated by new criterion method is more reliable? Moreover, you can highlight the REV region in Figure 4 so that readers can directly see that.

**[Response]** Comments accepted. The relative gradient error is proposed by previous study and has also used for our research about translucent porous media and contaminants migration. However, random fluctuations exist in $\varepsilon_g^i$ curves visually, which make the REV plateau uneasy to be identified. Significantly, the curve of new criterion appears low value interval which makes the beginning and ending of REV plateau easier to be identified. We have used open circles to indicate the REV plateau region in Fig. 4. Readers can see REV plateau estimated by the new criterion.

(6) Figure 6. There is not any interpret or discussion on the Figure 6. If the figure is important, please provide detail description. If not, I suggest moving it to the supplementary.

**[Response]** Comments accepted. Fig.7 is obtained on the REV distribution presented in Fig. 6. We have added more discussion about Fig.6 in revised manuscript (Lines 407-419).

(7) L383-384. In the downright corner of the Figure 7a, the red line increases first, then decrease. So I do not agree with that "while REV of PCE plume presents apparent decreasing ... for Experiment-II".

**[Response]** Comments accepted. We have revised this sentence in revised manuscript (Lines 447-448).

(1) L54, "As measured scale size ranging between Lmin and Lmax," Please give the Lmin and Lmax

directly in the figure.

**[Response]** Comments accepted. We have added "$L_{min}$" and "$L_{max}$" in Fig. 1a (Lines 683-685).

(2) Is there any reference for the conceptual representation of "REV curve" in L50?

**[Response]** Comments accepted. We have added reference for "REV curve" (Lines 52-53).

(3) L142. "Fig. 1c" should be "Fig. 1d".

**[Response]** Comments accepted. We have made corresponding correction (Line 152).

(4) L148. What does "n" mean in the Equation 5? And, the porosity does not occur in the Equation, how do you derive the porosity from it?

**[Response]** Comments accepted. We have replaced 'n' with '$\theta$' (Lines 53 and 158).

(5) L218-220. What is the difference between the and? Are they the same?

**[Response]** Comments accepted. We have corrected the sub and sup $i$ (Line 256).

(6) The author should proofread the paper carefully, as the current paper has numerous typos.
For example,
L243: "Figure 2c" cannot be found in the paper.
L358, "All mean REV sizes of these variables for Experiment-II are larger than REVs of Experiments-II".
L386-387, the sentence does not have verb.

**[Response]** Comments accepted. We have checked carefully and corrected these mistakes above (Lines 180, 402 and 431-433).

[revised manuscript text omitted]

*θ represents porosity, ρ represents density, τ represents tortuosity; ω, ϵ and ν are fitted parameters of the model

**Table 34.** The fitted equations between average value of REV and $d_I$, $d_m$

| Experiment | | $d_m$ | $d_I$ |
|---|---|---|---|
| I | $S_o$-REV | $\overline{REV} = -1.67 \times 10^{-3}d_m^2 + 0.193d_m + 2.72$ (R²=0.807) | $\overline{REV} = -1.97 \times 10^{-3}d_I^2 + 0.245d_I + 1.12$ (R²=0.832) |
| | $A_{ow}$-REV | $\overline{REV} = -6.10 \times 10^{-4}d_m^2 + 5.82 \times 10^{-2}d_m + 7.20$ (R²=0.401) | $\overline{REV} = -1.47 \times 10^{-3}d_I^2 + 0.205d_I + 1.84$ (R²=0.733) |
| II | $S_o$-REV | $\overline{REV} = -4.08 \times 10^{-5}d_m^2 + 1.50 \times 10^{-2}d_m + 7.54$ (R²=0.655) | $\overline{REV} = -3.94 \times 10^{-5}d_I^2 + 7.80 \times 10^{-3}d_I + 8.50$ (R²=0.327) |
| | $A_{ow}$-REV | $\overline{REV} = -1.92 \times 10^{-5}d_m^2 + 4.47 \times 10^{-3}d_m + 9.46$ (R²=0.616) | $\overline{REV} = -1.92 \times 10^{-5}d_I^2 + 4.47 \times 10^{-3}d_I + 9.46$ (R²=0.616) |
| III | $S_o$-REV | $\overline{REV} = -6.06 \times 10^{-6}d_m^2 + 2.27 \times 10^{-3}d_m + 7.76$ (R²=0.153) | $\overline{REV} = 1.69 \times 10^{-5}d_I^2 - 1.21 \times 10^{-2}d_I + 9.62$ (R²=0.236) |
| | $A_{ow}$-REV | $\overline{REV} = -8.71 \times 10^{-6}d_m^2 + 5.66 \times 10^{-3}d_m + 11.5$ (R²=0.115) | $\overline{REV} = -1.50 \times 10^{-5}d_I^2 + 7.88 \times 10^{-3}d_I + 11.4$ (R²=0.150) |

* $\overline{REV}$ is the average value of REV size, $d_m$ is the distance from mass center of PCE

plume to the cell contained in PCE plume, $d_I$ is the distance from injection point to the cell contained in PCE plume

**Table 4.** The fitted relationship between REV and $S_o$

 |  |
---|---|---
 |  |
 |  |

| Experiment | $S_o$-REV | $A_{OW}$-REV |
|---|---|---|
| I | $REV = -2.13 \times \ln S_o + 0.876$ ($R^2$=0.466) | $REV = 2.27e^{2.70 \times S_o}$ ($R^2$=0.366) |
| II | $REV = -0.961 \times \ln S_o + 1.09$ ($R^2$=0.415) | $REV = 1.70e^{3.30 \times S_o}$ ($R^2$=0.500) |
| III | $REV = -1.40 \times \ln S_o + 3.96$ ($R^2$=0.538) | $REV = 2.05e^{3.22 \times S_o}$ ($R^2$=0.573) |

**Figure Captions**

**Figure 1.**  (a) Variable changes as measured scale (L) increment in conceptual curve (Costanza-Robinson et al., 2011); (b) Scale effect and the cuboid image window geometry; (c) System Device for acquisition of parameters (porosity and density, etc.) of translucent material; (d) The infinitesimal selected from translucent porous media packed in 2D sandbox.

**Figure 2.** (a) The system sandbox equipment of Experiment-I; (b) The system sandbox equipment of Experiment-II; (c) The system sandbox equipment of Experiment-III

**Figure 3.** (a) The emergent light intensity, porosity, permeability and tortuosity of 2D translucent silica sand for Experiments-I-III ; (b) The PCE saturation of Experiments-I-III  during 0~1523 min and observation cells

**Figure 4.** (a) The change of porosity (θ), associated coefficient of variation ($C_V^i$ ), entropy dimension ($DI^i$ ), the relative gradient error ($\varepsilon_g^i$ ), and new criterion-$\chi^i$ for observation cells as cuboid window scale (L) increases; (b) The change of PCE saturation ($S_o$), associated $C_V^i$ , $DI^i$ , $\varepsilon_g^i$ , and $\chi^i$ for observation cells as cuboid window scale (L) increases

**Figure 5.** (a) The distributions of minimum REV sizes of porosity, sand density and tortuosity for Experiments-I-III ; (b) The frequency of minimum REV sizes of Experiments and fitted models

**Figure 6.** (a) The distributions of $S_o$-REV sizes during 0~1523 min for Experiments-I-III

; (b) The distributions of $A_{OW}$-REV sizes during 0~1523 min for Experiments-

I-III

**Figure 7.** (a) The mass center coordinate of PCE plume, GTP, plume area and the mean, standard deviation of $S_o$-REV and $A_{OW}$-REV during 0~1523 min

; (b)

~~0~1523 min~~ The change of average REV size as the distance $d_I$, $d_m$ increases and fitted relationship between REV sizes and $S_o$ for Experiments-I and II

**Fig. 1**

(a)

[Figure]

(b)

[Figure]

(c)
[Figure]
 (d)

[Figure]

(a)

(b)

[Figure]

(c)

[Figure]

(d)

**Fig. 2**

[Figure]

[Figure]

**Fig. 3**

[Figure]

[Figure]

**Fig. 4**

[Figure]

[Figure]

**Fig. 5**

[Figure]

[Figure]

**Fig. 6**

[Figure]

[Figure]

**Fig. 7**

[Figure]

[Figure]

---

## Author Response (AR2)

Response to Editor's and Referees' Comments on Manuscript hess-2020-91 "A new criterion for determining the representative elementary volume of translucent porous media and inner contaminant" by Ming Wu, Jianfeng Wu, Jichun Wu, and Bill X. Hu

Note that the following text in Arial Narrow font denotes Editor's and Reviewers' comments and in Times New Roman font denotes our response to the comments in the review. In our Supplement (pdf/zip), the PDF file has clearly indicated all changes to the original manuscript. In our Manuscript (pdf), all changes have been included in the clean version of the revised PDF file. Also, in our marked PDF file, marked in  is the text that should be removed from the original manuscript and marked in a red font is the text that has been added to the revision. In addition, Line number(s) mentioned below is referred to as that line numbering in the marked revised manuscript.

**Response to Anonymous Referees' Comments**

We received two review reports submitted by two anonymous referees. Report #1 from Referee #3 suggested the manuscript should be "accepted as is"; Report #2 submitted by Referee #1 suggested the manuscript should be "accepted subject to minor revisions". Suggestions for revision are as follows:

(1) Line 44, references are suggested to add before "Previous studies... ".

**[Response]** References are added after "Previous studies" (Line 44).

(2) Line 590, Table 2 represents the parameter, not equations! Title should be revised.

**[Response]** We have revised the title of Table 2 as the referee suggested (Lines 592-593).

[revised manuscript text omitted]

**Figure Captions**

**Figure 1.** (a) Variable changes as measured scale (L) increment in conceptual curve (Costanza-Robinson et al., 2011); (b) Scale effect and the cuboid image window geometry; (c) System Device for acquisition of parameters (porosity and density, etc.) of translucent material; (d) The infinitesimal selected from translucent porous media packed in 2D sandbox.

**Figure 2.** (a) The system sandbox equipment of Experiment-I; (b) The system sandbox equipment of Experiment-II; (c) The system sandbox equipment of Experiment-III

**Figure 3.** (a) The emergent light intensity, porosity, permeability and tortuosity of 2D

translucent silica sand for Experiments-I-III; (b) The PCE saturation of

Experiments-I-III during 0~1523 min and observation cells

**Figure 4.** (a) The change of porosity (θ), associated coefficient of variation ($C_V^i$), entropy dimension ($DI^i$), the relative gradient error ($\varepsilon_g^i$), and new criterion-$\chi^i$ for observation cells as cuboid window scale (L) increases; (b) The change of PCE saturation (S$_o$), associated $C_V^i$, $DI^i$, $\varepsilon_g^i$, and $\chi^i$ for observation cells as cuboid window scale (L)

increases

**Figure 5.** (a) The distributions of minimum REV sizes of porosity, sand density and tortuosity for Experiments-I-III; (b) The frequency of minimum REV sizes of

Experiments and fitted models

**Figure 6.** (a) The distributions of S$_o$-REV sizes during 0~1523 min for Experiments-I-III; (b) The distributions of A$_{OW}$-REV sizes during 0~1523 min for Experiments-I-III

**Figure 7.** (a) The mass center coordinate of PCE plume, GTP, plume area and the mean, standard deviation of S$_o$-REV and A$_{OW}$-REV during 0~1523 min; (b) The change of average REV size as the distance $d_I$, $d_m$ increases and fitted relationship between

REV sizes and $S_o$ for Experiments-I and II

**Fig. 1**

[Figure]

[Figure]

[Figure]

**Fig. 2**

(a) Experiment-I

(a) Experiment-I

outflow

Effluent

E-1
E-2

cm

E-3

Injection
Point

**F40/F50**

Uniform flow cm

I-1
I-2

I-3

Inflow

Peristaltic
pump

Influent (b) Experiment-II

outflow

Effluent

E-4

E-5

cm

E-6

Sample
Needle

Injection
Point

**F20/F30**

Uniform flow cm

I-4

I-5

I-6

Inflow

Peristaltic
pump

Influent (c) Experiment-III

outflow

Effluent

E-7

E-8

cm

E-9

A(F70/F100)

B(F40/F60)          C(F70/F80)

D(F50/F70)

E(F30/F40)

**F20/F30**

Uniform flow cm

I-7

I-8

I-9

Inflow

Peristaltic
pump

Influent

**Fig. 3**

[Figure]

[Figure]

**Fig. 5**

[Figure]

**Fig. 6**

[Figure]

**Fig. 7**

[Figure]